# Ion desolvation for boosting the charge storage performance in Ti$_3$C$_2$ MXene electrode

Zheng Bo [1], Rui Wang[1], Bin Wang[2], Sanjay Sunny[3], Yuping Zhao[4], Kangkang Ge [3], Kui Xu [5], Yajing Song[4], Encarnacion Raymundo-Piñero[6], Zifeng Lin [2], Hui Shao [7], Qian Yu [4] ✉, Jianhua Yan[1] ✉, Kefa Cen[1], Pierre-Louis Taberna [3,8] & Patrice Simon [3,8] ✉

Clarifying the relationship between ion desolvation, ion-electrode interactions, and charge storage capacity during ion intercalation in host electrode materials is crucial for advancing fast and efficient energy storage systems. However, the absence of direct evidence for ion desolvation and lack of detailed understanding of the interactions between surface terminations and intercalated cations (Li ions)/solvents hinder the exploration of their effects on energy storage mechanisms. In this paper, we study the intercalation of Li ions from a non-aqueous electrolyte in two-dimensional metal carbides Ti$_3$C$_2$ MXenes with different surface chemistries: HF-Ti$_3$C$_2$ (F-, OH- and O-terminated) and MS-Ti$_3$C$_2$ (O- and Cl-terminated) MXenes. We are able to visualize the full ion desolvation and solvents-ions co-intercalation in the interlayers of MS-MXene and HF-MXene, respectively at the atomic scale. The combination of several techniques and characterization tools reveal that the complete ion desolvation in Cl- and O-terminated MS-Ti$_3$C$_2$ MXenes is associated with the formation of a dense solid electrolyte interface layer, resulting in improved charge storage capacity. The O-rich surface terminations of MS-MXenes are found to be responsible for the efficient Li ions storage. These findings shed lights on identifying the critical role of non-electrostatic ion-electrode interactions and ion desolvation in designing high-performance energy storage devices.

The understanding of electrolyte ion dynamics at the electrolyte/electrode interface as well as the organization of ions in the bulk of active material, is of key importance for improving the charge storage in battery electrodes[1–6]. The degree of interaction between ions and the host material is highly related to the charge storage mechanism, with mainly two processes clearly identified: (1) purely electrostatic, non-Faradaic electrosorption of fully solvated ions forming the electric double layer, known as the capacitive storage, and (2) a purely Faradaic

[1]State Key Laboratory of Clean Energy Utilization, College of Energy Engineering, Zhejiang University, Hangzhou, China. [2]College of Materials Science and Engineering, Sichuan University, Chengdu, China. [3]CIRIMAT UMR CNRS 5085, Université de Toulouse, Toulouse, France. [4]Center of Electron Microscopy and State Key Laboratory of Silicon and Advanced Semiconductor Materials, School of Materials Science and Engineering, Zhejiang University, Hangzhou, China. [5]School of Flexible Electronics (Future Technologies) & Institute of Advanced Materials (IAM), Nanjing Tech University, Nanjing, China. [6]CNRS CEMHTI UPR3079, Université Orléans, Orléans, France. [7]i-Lab, CAS Center for Excellence in Nanoscience, Suzhou Institute of Nano-Tech and Nano-Bionics (SINANO), Chinese Academy of Sciences (CAS), Suzhou, China. [8]FR CNRS 3459, Réseau sur le Stockage Electrochimique de l'Energie (RS2E), Amiens, France. ✉e-mail: yu_qian@zju.edu.cn; yanjh@zju.edu.cn; patrice.simon@univ-tlse3.fr

process with full charge transfer to the host material following by the intercalation of fully desolvated ions such as Li, Na, K..., occurring in the last generation of metal-ion battery electrodes[7,8]. Those different charge storage processes come with key different features: high power and low energy density for capacitive storage (electrostatic, surface process) and high energy and limited power density for batteries (charge transfer, bulk process). When the electrosorption of ions is achieved in nanoconfined spaces (nm or less), a continuous transition exists between these two extreme cases, where the charge storage is determined by the extent of ion (de)solvation and ion/host interaction[7]. This is the region where some of the so-called "pseudo-capacitive" processes are observed. The more the ion desolvation, the closer it is to the surface of the host material and the stronger the ion/host interaction, resulting in partial charge transfer and increased charge density. It is similar to our understanding of chemical bonding, which is rarely an "ideal" primary bonding type such as covalent or ionic but more iono-covalent, or electrodes, which are rarely ideally non-polarizable (Faradaic) or polarizable (non-Faradaic)[7]. Thus, figuring out the solvent-ions interaction and ions-electrode interaction upon the processes of adsorption, intercalation, extraction and transportation of ions in host materials is overarching to probing the energy storage mechanism[2,4,9–12].

Thanks to its metal-like conductivity, 2-D structure, tunable surface chemistry, and flexible interlayer space[13–16], MXenes, a family of two-dimensional transition metal carbides and nitrides, have been used as model materials for probing the energy storage mechanisms[17–20]. MXenes are usually prepared by etching the "A" element of MAX phase (A = early transition metal) in HF-containing electrolytes, resulting in the preparation of F-, OH-, and O-terminated MXenes[21–23]. Previous theoretical studies have shown the dominant role of the surface chemistry of HF-MXenes on lithium ions storage mechanism[24–27]. Notably, first-principles calculations predicted that the desolvation of cations would enhance their interaction with surface terminations in MXenes and charge transfer, resulting in improved energy storage performance[28]. In 2019, the intercalation of Li ions in the confined interlayer space of F-terminated $Ti_3C_2$ MXene was reported, resulting in an increased capacity by fast redox charge transfer[20]. The partial desolvation of the Li ion when entering the MXene structure was proposed to be at the origin of the capacity improvement. More recently, Lewis acid melts were reported as an alternative synthetic route for MXenes, termed as MS-MXenes, which circumvents the use of hazardous F-containing etching electrolytes and broadens the family of MXenes and their surface chemistry[29–31]. O- and Cl-terminated MS-$Ti_3C_2$ MXene exhibited high capacity (beyond 200 mAh g$^{-1}$) and high-power performance when used as the negative electrode during Li ion intercalation in the carbonate-based electrolyte, while conventional HF etched $Ti_3C_2$ MXene exhibited a poor electrochemical behavior in the same condition[30,31]. In-situ-XRD results suggested that different evolutions of $d$-spacing of two MXenes might be contributed to the different degrees of Li ion desolvation, which plays a significant role in pseudocapacitive processes. However, there is still a lack of direct evidence of this desolvation resulting in the ambiguity of how the terminations affect the ion desolvation. The reasons for this lie in the difficulty of obtaining an atomic resolution view of surface terminations, intercalated cations (Li ions) and solvents, as well as the complexity of surface chemistry/solvent interactions, all of which indicate a lack of understanding of how surface terminations contribute to energy storage and the correlation between ion (de)solvation and charge storage mechanisms[17,32].

In this work, we used high-resolution scanning transmission electron microscopy coupled with integrated differential phase contrast (iDPC-STEM) technique to characterize MS- and HF-$Ti_3C_2$ MXene during Li intercalation in a non-aqueous carbonate-based electrolyte. Li ion desolvation and solvents co-intercalation in the MXene interlayers were clearly visualized at the atomic scale for the first time in Cl-

and O-terminated MS-$Ti_3C_2$ and conventional F-, OH- and O-terminated HF-$Ti_3C_2$, respectively. $^7$Li solid-state nuclear magnetic resonance (ssNMR) measurements and thermo-programmed desorption coupled with mass spectroscopy experiments (TPD-MS) confirmed the presence of solvent molecules confined between the $Ti_3C_2$ layers in HF-MXene samples, while almost no solvent was found in MS-MXene. Moreover, a dense solid electrolyte interphase (SEI) layer was formed on the surface of MS-$Ti_3C_2$ with sufficient −O terminations which might facilitate the Li ions desolvation when entering the MXene interlayer spacing, resulting in a capacity increase.

## Results

MS-$Ti_3C_2T_x$ MXenes were prepared by etching Al element of $Ti_3AlC_2$ MAX phase at 750 °C in a molten salt mixture of KCl, LiCl, and $CuCl_2$ as etchants. The MXene powder was further washed with ammonium persulfate (APS) to remove Cu dots (see "Methods") to obtain the final MS-$Ti_3C_2T_x$ samples, where $T_x$ stands for O- and Cl-terminations[30]. A second series of samples, namely HF-$Ti_3C_2T_x$ MXenes, was obtained from the etching of $Ti_3AlC_2$ MAX phase in LiF + HCl electrolyte at room temperature, with a surface chemistry $T_x$ ($x$ = −O, −OH, and −F) different from the MS-$Ti_3C_2T_x$ MXenes[22,23,33].

Figure 1a, b show the cross-section high-angle annular dark-field scanning transmission electron microscopy (HAADF-STEM) images of both pristine MS-$Ti_3C_2T_x$ and HF-$Ti_3C_2T_x$ MXene slices cut by focused ion beam (FIB) technique, exhibiting the expected layered structures. FIB milling causes layer bending in certain areas. The HAADF-STEM images at higher magnification show the atomic-scale layered structure of both MS-$Ti_3C_2T_x$ and HF-$Ti_3C_2T_x$ MXenes in Fig. 1c, d, which contains three layers of titanium (Ti) atoms along the [11$\bar{2}$0] direction per unit. Since the contrast of common HAADF-STEM technology is related to the square of the atomic number Z (-Z$^2$), light elements could be barely visible using this technique. Advanced iDPC-STEM technique was therefore employed, where the contrast is proportional to Z, drastically improving the simultaneous detection of light (e.g., C, O, F, and Li) and heavy elements (e.g., Ti and Cl)[34,35]. The iDPC-STEM image of MS-$Ti_3C_2T_x$ MXene in Fig. 1e now shows clear image of light elements (i.e., −O) of MS-$Ti_3C_2$'s terminations, which could be hardly reached in the conventional HAADF mode (Fig. 1c). Besides, the $d$-spacing of MS-$Ti_3C_2$ is clearly visible, reaching about 1.1 nm. The O-terminations are connected on the surface Ti layers with the same structure. The iDPC-STEM image of HF-$Ti_3C_2T_x$ in Fig. 1f shows the same atomic structure as O-terminated MS-$Ti_3C_2T_x$ layers with a different $d$-spacing value (around 1.3 nm).

Electron energy loss spectroscopy (EELS) analysis[36] was conducted on pristine MS-$Ti_3C_2T_x$, in which O signals (at around 531 eV) and weak Cl signals (at around 207 eV) were found in the spectra shown in Fig. 1g. TPD-MS was performed on MS-$Ti_3C_2T_x$ and HF-$Ti_3C_2T_x$ samples to analyze the O-containing terminations of the two $Ti_3C_2T_x$ samples, and the results are shown in Supplementary Fig. 1. Based on X-ray photoelectron spectroscopy (XPS) depth profiling and energy dispersive X-ray spectroscopy (EDS) analyses (Supplementary Table 1 and Supplementary Fig. 2), it is found that the −O is the main surface terminations of MS-$Ti_3C_2$. Few regions of MS-$Ti_3C_2T_x$ show external two layers of Cl atoms on the top of the surface Ti layers, as shown in Fig. 1h, and −OH terminations are rarely found, in agreement with previous results[30]. The low Cl-termination content in MS-MXene is explained by slightly different sample preparation conditions (see "Methods")[30].

Different surface chemistry was measured for the HF-$Ti_3C_2$ sample, as −O, −OH, and −F surface terminations were present, in agreement with the literatures[22,23,33]. The content of O-terminations in MS-$Ti_3C_2T_x$ is higher than that of HF-$Ti_3C_2T_x$ (see Supplementary Figs. 1 and 2) due to the final oxidation treatment to remove Cu. The values of $d$-spacing of MS-$Ti_3C_2T_x$ and HF-$Ti_3C_2T_x$ are calculated as 11.0 Å and 13.0 Å, respectively from X-ray diffraction (XRD) patterns

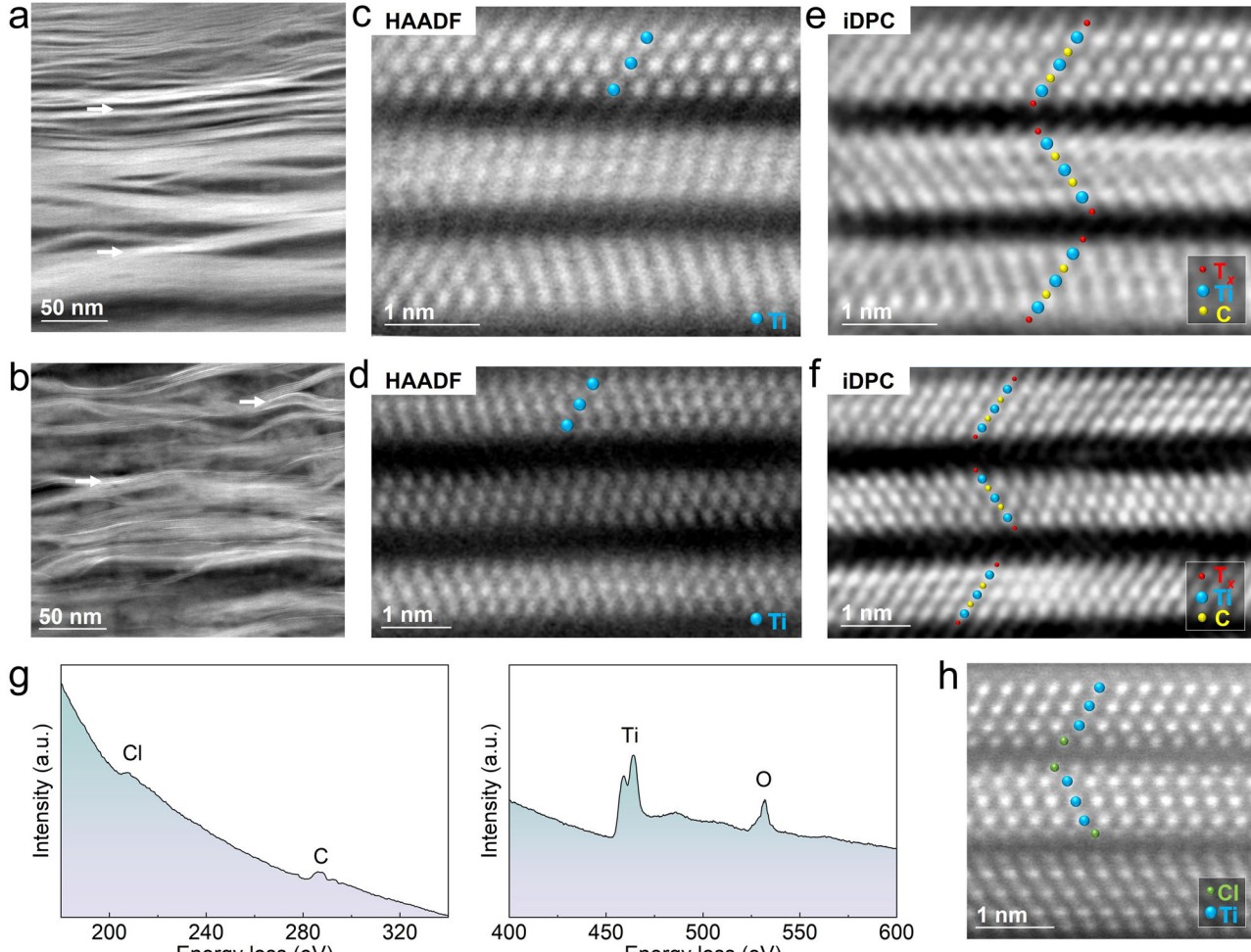

**Fig. 1 | The atomic structure of MS-Ti$_3$C$_2$T$_x$ and HF-Ti$_3$C$_2$T$_x$.** The HAADF-STEM images with low magnification of the FIB milling slices of (**a**) pristine MS-MXene and (**b**) pristine HF-MXene. The bright white ribbons marked by white arrows are the regions with the appropriate zone axis which could be imaged in iDPC-STEM mode. **c** The HAADF-STEM image of layer-structured MS-Ti$_3$C$_2$T$_x$ with the *d*-spacing of 1.1 nm. Ti atoms are marked by blue dots. **d** The HAADF-STEM image of layer-structured HF-Ti$_3$C$_2$T$_x$ with the *d*-spacing of 1.3 nm. Ti atoms are marked by blue dots. **e** The iDPC-STEM image of MS-Ti$_3$C$_2$T$_x$ which is corresponding to (**c**). Ti atoms, C atoms, and surface terminations are marked by blue dots, yellow dots, and red dots, respectively. **f** The iDPC-STEM image of HF-Ti$_3$C$_2$T$_x$ which is corresponding to (**d**). Ti atoms, C atoms, and surface terminations are marked by blue dots, yellow dots, and red dots, respectively. **g** The EELS spectra of MS-Ti$_3$C$_2$T$_x$ indicating the existence of Ti, C, O, and Cl signals. **h** HAADF-STEM image of the region terminated with −Cl in MS-Ti$_3$C$_2$T$_x$. Ti atoms and Cl atoms are marked by blue and green dots, respectively. Source data are provided as a Source Data file.

(see Supplementary Fig. 3), consistent with the *d*-spacing measured in iDPC-STEM images.

Electrochemical characterizations were achieved in two-electrode coin cells (half cells) using MS-Ti$_3$C$_2$T$_x$ and HF-Ti$_3$C$_2$T$_x$ as working electrodes and Li foil as a counter electrode, in 1 M LiPF$_6$ (in 1:1 vol/vol ethylene carbonate/dimethyl carbonate, LP30) electrolyte. Cyclic voltammograms (CV) and galvanostatic charge-discharge (GCD) experiments were achieved within a 0.2–3 V vs Li$^+$/Li voltage window, as shown in Fig. 2a and Supplementary Figs. 4−6. While a mirror-like electrochemical signature is observed for MS-Ti$_3$C$_2$, suggesting fast, reversible Li ion intercalation/deintercalation in the host MS-Ti$_3$C$_2$T$_x$, HF-Ti$_3$C$_2$T$_x$ shows pairs of redox peaks, suggesting sequential Li intercalation in slits[30]. The measured discharge capacity of MS-Ti$_3$C$_2$T$_x$ is 873 C g$^{-1}$ at the scan rate of 0.1 mV s$^{-1}$, that is almost three times of HF-Ti$_3$C$_2$T$_x$ (286 C g$^{-1}$). Moreover, the *b* value obtained from Eq. 2 (see "Methods") for MS-Ti$_3$C$_2$T$_x$ is larger than that of HF-Ti$_3$C$_2$T$_x$ (of 0.912 vs 0.709, respectively) in the range of 0.5 to 100 mV s$^{-1}$, indicating faster ion transport in MS-Ti$_3$C$_2$T$_x$ as shown in Fig. 2b[8]. Notably, the low Coulombic efficiency (CE) and irreversible capacity loss at the first

cycle suggest the formation of SEI layer for both MXene samples, as shown in Supplementary Fig. 4[37].

These results are consistent with previous results by Liu et al., who studied the electrochemical behavior of O-free, Cl-terminated Ti$_2$CCl$_x$ MXene prepared from electrochemical etching in molten salt electrolytes[38]. They showed that O-free, Cl-terminated Ti$_2$CCl$_x$ MXene exhibited poor electrochemical performance with low capacity and power capability[38]. Differently, after oxidation in APS, Cl- and O-terminated Ti$_2$CCl$_x$O$_y$ MXene could achieve a similar electrochemical signature and excellent performance as those of MS-MXenes reported here, with high capacity (860 C g$^{-1}$) and power density. The increase in oxygen content was then proposed to be the origin of the drastic change in the electrochemical behavior and performance[38].

In-situ-XRD measurements were conducted to depict the real-time evolution of the interlayer spacing of MXenes, by tracking the 002 peak position. The *d*-spacing stays constant at 11.0 Å during the cycling for MS-Ti$_3$C$_2$T$_x$ (see Fig. 2c). Differently, the *d*-spacing of HF-Ti$_3$C$_2$T$_x$ varies between 13.7 and 14.9 Å during cycling (see Fig. 2d). Such change in the *d*-spacing suggests the different charge storage

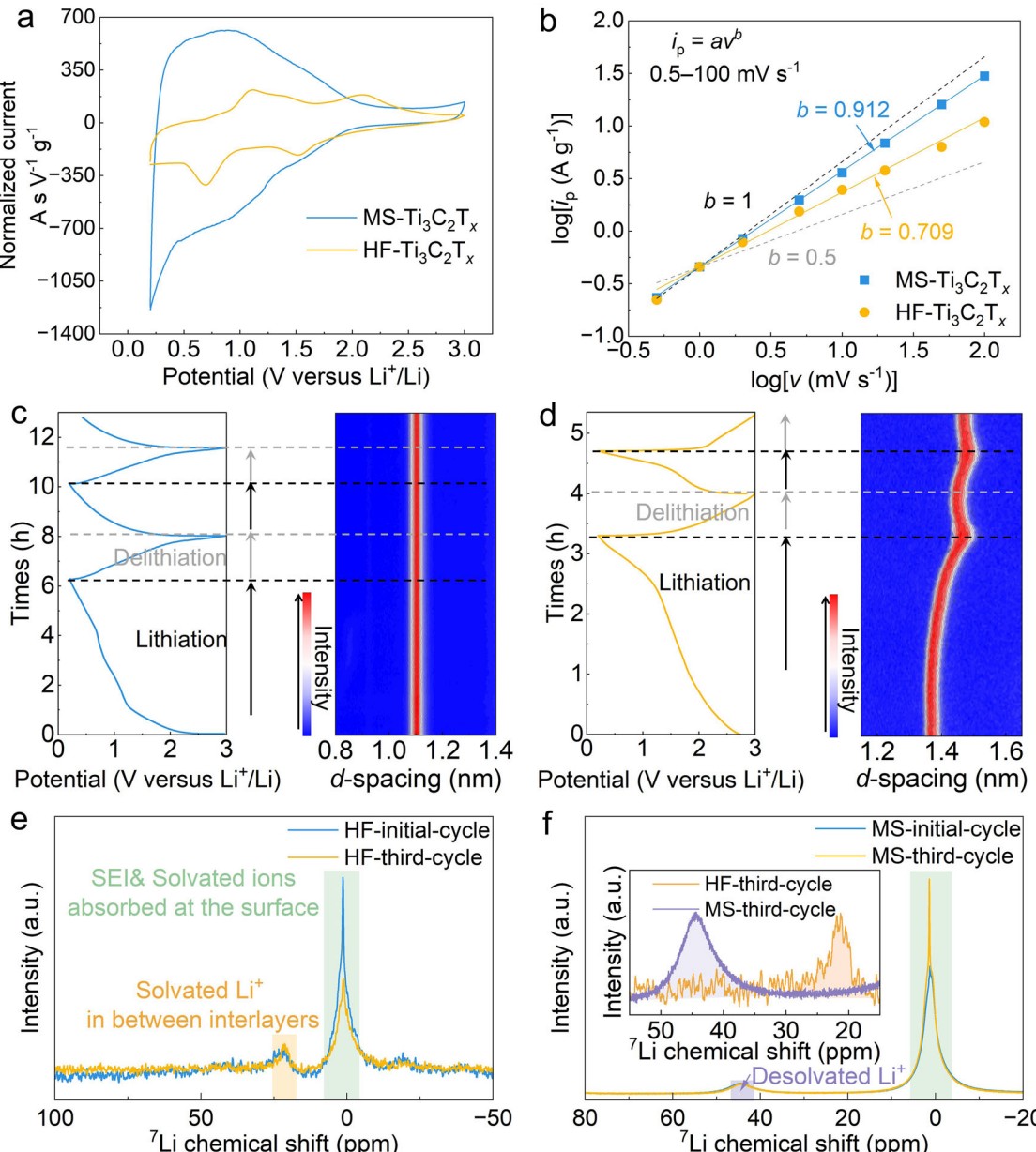

**Fig. 2 | The electrochemical properties of MS-Ti₃C₂Tₓ and HF-Ti₃C₂Tₓ. a** CV tests of MS-Ti₃C₂Tₓ and HF-Ti₃C₂Tₓ at the scan rate of 0.1 mV s⁻¹ in 1 M LiPF₆ (in 1:1 ethylene carbonate/dimethyl carbonate) electrolyte. **b** The $b$ values of MS-Ti₃C₂Tₓ (0.912 ± 0.003) and HF-Ti₃C₂Tₓ (0.709 ± 0.024) in the range of 0.5 to 100 mV s⁻¹. **c** In-situ-XRD maps of (002) peaks of MS-Ti₃C₂Tₓ during GCD tests. **d** In-situ-XRD maps of (002) peaks of HF-Ti₃C₂Tₓ during GCD tests. ⁷Li ssNMR spectra of (**e**) HF-

Ti₃C₂ samples lithiated in the initial and third cycle, and (**f**) MS-Ti₃C₂ samples lithiated in the initial and third cycle indicating the Li ions confined in between interlayers and free Li ions in SEI layers and absorbed at surface. The inset shows the comparison between the integrated areas of the peaks for intercalated Li ions in MS-Ti₃C₂ (purple region) and HF-Ti₃C₂ (orange region), illustrating more Li ions inserted in MS-Ti₃C₂ than HF-Ti₃C₂. Source data are provided as a Source Data file.

mechanisms for both samples. For HF-Ti₃C₂Tₓ, the $d$-spacing begins to increase below 1.8 V, which is consistent with the first cathodic peak corresponding to the beginning of Li ions insertion (Fig. 2a). The increase in the $d$-spacing could be consistent with solvent co-intercalation with Li ions, which expands the HF-Ti₃C₂Tₓ interlayers. A slight decrease in $d$-spacing is observed during the delithiation process (oxidation), up to 3 V. Importantly, the 002 peak position does not return to its original value, and this irreversible change persists in the subsequent cycle. These findings suggest that solvated Li ion insertion is achieved during the first charge (reduction). An excess of solvent molecules stays trapped between HF-Ti₃C₂Tₓ interlayers after the first cycle, indicating different Li ions-MXene interactions.

⁷Li ssNMR spectroscopy experiments[39,40] were made to further probe the chemical environment of Li ions storage during the lithiation

and delithiation. HF- and MS-MXene samples were prepared by polarization at a constant potential of 0.2 V (vs. Li⁺/Li) after the initial (first) and after three CV cycles at 0.5 mV s⁻¹. The lithiated electrodes were then washed by DMC to remove the LP30 electrolytes at the surface, and the same weight of materials was introduced in the rotors for NMR experiments. The XRD measurements shown in Supplementary Fig. 7a confirm that washing with DMC has a negligible effect on the solvents accommodated in the interlayers. These samples are further referred to as MS-initial-cycle, MS-third-cycle, HF-initial-cycle, and HF-third-MXene, respectively. The ssNMR spectra in Fig. 2e, f show the presence of a sharp peak at around 0 ppm which is attributed to solvated lithium ions adsorbed at the MXene surface and/or present in the SEI layer[41]. Besides, the additional peaks around 22 ppm and 45 ppm that appear in the lithiated samples but not in pristine MXenes

powders (Supplementary Fig. 7b) are assigned to the presence of intercalated Li ions in the interlayers of MXenes[42]. As the NMR probes the chemical states of both lithium and MXene, along with their interactions, the shift of the intercalated Li ions peak from 22 ppm to 45 ppm indicates the stronger confinement effect of Li ions in MS-MXene than that in HF-MXene[43,44]. Such a shift is compatible with the presence of Li ions in the interlayers of HF-MXene with more solvent molecules, resulting in decreased interaction with the electrode surface, compared to MS-MXene. Moreover, the area of peak (a.u.) corresponding to Li signals in SEI layer is much higher in MS-MXene than that in HF-MXene, which could be explained by the presence of a thicker SEI layer formed at the MS-MXene/electrolyte interface. After cycling, the sharply decreased area of the peak around 0 ppm could support the degradation of the SEI layer formed on HF-MXene (see below). It's also worth noting that the integrated area of intercalated Li ions in MS-third-cycle sample (purple region) is larger than that of HF-third-cycle sample (orange region) as shown in the inset of Fig. 2f, which illustrates that ion desolvation could lead to more Li ions inserted into the interlayers of MS-MXene, contributing to the improved capacity, even though the $d$-spacing of MS-MXene is slightly smaller than that of HF-MXene.

As the interaction between Li ions and solvents and the structure at the MXene/electrolyte interface are distinct in MS- and HF-MXene, further investigations were made on the arrangement of Li ions and solvent molecules in lithiated MXene electrodes by techniques including iDPC-STEM, HAADF-STEM, and EELS analyses. All samples were prepared by electrochemical polarization at a constant potential of 0.2 V (vs. Li$^+$/Li) during the initial stage and after three CV cycles. The TEM samples need to be ion-milled to ultra-thin slices to minimize the projection effect. The low magnification cross-section HAADF-STEM images of four lithiated MXene FIB samples are showed in Supplementary Fig. 8, exhibiting the layered structure. The EDS mapping and spectrum of lithiated MS-MXene FIB milling sample demonstrate the negligible effect of Gallium (Ga) ions on the samples during FIB cutting in Supplementary Fig. 9. Figure 3a, b illustrate the arrangement of Li ions and solvents in the HAADF-STEM and the corresponding iDPC-STEM mode of MS-Ti$_3$C$_2$T$_x$ after lithiation, respectively. The additional individual white dots located in the middle of interlayers, highlighted by a purple dashed frame, are clearly resolved in iDPC-STEM images but cannot be seen in the corresponding HAADF-STEM images in Fig. 3a, b. The HAADF-STEM and corresponding iDPC-STEM images in other larger regions of lithiated samples are showcased in Supplementary Fig. 10. Moreover, intercalated Li ions are clearly shown in the iDPC-STEM images while they are absent in the pristine MS-Ti$_3$C$_2$T$_x$ (Fig. 1e). It is worth noting that although the extra dots in the interlayer spacing in Fig. 3a, b may not all correspond to lithium atoms individually, the majority of the contrast likely originating from Li ions is plausible since it is quite weak and only appear in iDPC-STEM mode. In Fig. 3c and Supplementary Fig. 11a, the presence of O and Li signals and weak C and Cl signals in the EELS-STEM mapping indicates the almost complete desolvation of Li ions between the interlayers of MS-Ti$_3$C$_2$T$_x$ MXene. This is further verified by the O-Ti-C-Ti-C-O-Li unit in the intensity profiles in the regions where the terminations are invisible in the HAADF-STEM image, as indicated by the white arrow (Supplementary Fig. 11b). These results are consistent with theoretical modeling studies reporting that Li-ion storage capacity was strongly dependent on the surface terminations, with O-terminated Ti$_3$C$_2$ MXenes showing the highest storage capacities thanks to lower adsorption energy of Li ion on O-functionalities ($-1.404$ eV/Li atom)[26]. It is noted that the EELS signals are quite diffused at such high resolution, however the boundary of the signal across the interface is obvious. Importantly, no solvent molecules are detected in the interlayers of the MS-MXene (almost no carbon signal in the interlayers), which is consistent with the constant $d$-spacing (~11 Å) measured during cycling in Fig. 2c. The combined analysis from iDPC-STEM, HAADF-

STEM, intensity profiles, and EELS supports the intercalation of Li ions in O-terminated MS-Ti$_3$C$_2$ MXene by stripping off their solvation shell, leading to the presence of fully desolvated Li ions between the Ti$_3$C$_2$ layers. Additionally, Li ions are found on the surface of MS-Ti$_3$C$_2$T$_x$, interacting strongly with the $-$O terminations as illustrated in Fig. 3d.

Figure 3e, f show the local atomic arrangement of HF-Ti$_3$C$_2$T$_x$ (with T$_x$ = $-$O, $-$OH, and $-$F) sample after Li intercalation. A striking difference from the previous MS-Ti$_3$C$_2$T$_x$ is the increase of the $d$-spacing distance when compared to the pristine one, as can be seen in Fig. 3e (HF-initial-lithiation, $d$-spacing ≈ 14.5 Å) and Fig. 3f (cycled-HF-MXene, $d$-spacing ≈ 15 Å), which is in line with the in-situ-XRD measurements. Importantly, the presence of solvent molecules between HF-Ti$_3$C$_2$T$_x$ interlayers (noted as irregular white blocks in Fig. 3e, f) is now clearly visible from iDPC-STEM images, while the conventional HAADF-STEM falls short in detecting these light elements. The HAADF-STEM and corresponding iDPC-STEM images of continuous multi-layer regions (Supplementary Fig. 12a, b) demonstrated consistent $d$-spacing and ionic structures. The interlayer ionic structure is further established by mapping the distribution of Ti atoms, C atoms, O atoms, and Li ions (see Fig. 3g and Supplementary Fig. 12c) by EELS. The signals corresponding to Li and O (and C) indicate Li ions and solvent molecules, respectively appear throughout the interlayers. The corresponding EELS spectra images also support the existence of Li signals, and ion desolvation in MS-MXene and the ions-solvents co-intercalation in HF-MXene as shown in Supplementary Fig. 13. To summarize, Li ion intercalation in HF-Ti$_3$C$_2$T$_x$ MXene (T$_x$ = $-$O, $-$OH, and $-$F) is accompanied by solvent molecules (Fig. 3h), resulting in an increase of the interlayer distance, compared to pristine one. This process is then different from that of MS-Ti$_3$C$_2$T$_x$ MXene (T$_x$ = $-$O and $-$Cl), where fully desolvated Li ion intercalation is observed (Fig. 3d). It should be noted that the schematic does not indicate the real orientation and amount of the solvent molecules here. Since the main difference between the two MXene samples is the surface terminations, this suggests that the surface chemistry drives the Li ion desolvation and the charge storage mechanism.

TPD-MS experiments were performed to further study the insertion of solvents in MXene[45]. Here, MS-Ti$_3$C$_2$ and HF-Ti$_3$C$_2$ electrodes were polarized at 1.1 V (vs. Li$^+$/Li) until a steady state was reached (constant current). This potential was selected as it corresponds to the beginning of the (002) peak shifting on the HF-MXene, still being high enough to minimize the formation of the SEI layer on the electrode surface. The polarized electrodes were disassembled and dried in a glovebox, followed by vacuum drying at 80 °C for 12 h. TPD-MS was conducted on the MXenes before and after the vacuum drying which provided an insight into the surface adsorbed and the intercalated DMC (m/z = 45) as seen in Fig. 4a–d.

A broad and intense peak is observed between 130 and 200 °C for both MXene samples prior to vacuum drying, which disappears after drying (Fig. 4a, b). Removal of these peaks during the drying procedure further confirms the assignment to free and surface-adsorbed solvent molecules. In contrast, after vacuum drying, a significant peak is visible for HF-Ti$_3$C$_2$T$_x$ at higher temperature (200–250 °C) in Fig. 4a, b, which also appears as a shoulder peak before drying and is absent for pristine MXene powders (Fig. 4c). This new peak shifted to a higher temperature confirms that the solvent is evolving from a different site or trapped state[46]. Insertion of the solvated ions within the interlayer of MXene can lead to irreversible structural change and also lead to its trapping[47]. The absence of the peak for the dried MS-Ti$_3$C$_2$ (Fig. 4b) in the same temperature range indicates that Li ions are entering the interlayer desolvated. Note that the small peak observed for the MS-MXene sample in the range of 300 °C (green line in Fig. 4d) is also present in the pristine MS-MXene powder, which is not placed in contact with the electrolyte. This confirms that the observed peak is associated with the surface terminations of the MS-Ti$_3$C$_2$ rather than the solvents: the oxidation of carbon occurs during the APS washing

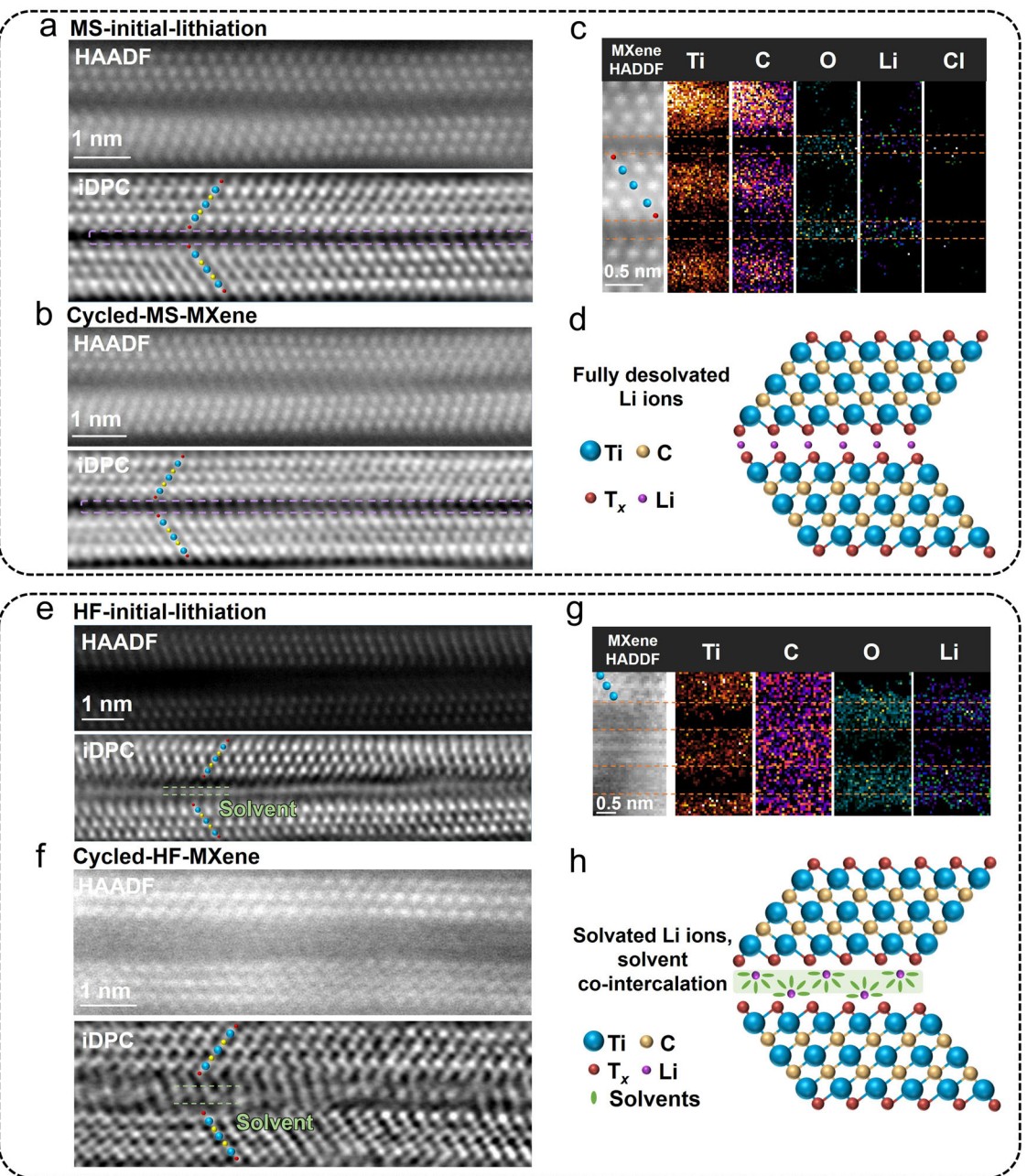

**Fig. 3 | Atomic-scale characterizations of ionic structures after electrochemical polarization.** The HAADF-STEM and corresponding iDPC-STEM images of (**a**) MS-initial-lithiation and (**b**) cycled-MS-MXene. Ti atoms, C atoms, and surface terminations are marked by blue dots, yellow dots, and red dots, respectively. The contrast of the terminations in MS-MXene is weak in HAADF images, but becomes clear in the corresponding iDPC images. **c** The EELS mapping of Ti, C, O, and Li of MS-initial-lithiation sample. Black represents the lowest intensity. **d** A schematic showing the full desolvation in MS-$Ti_3C_2T_x$. The HAADF-STEM and corresponding iDPC-STEM images of (**e**) HF-initial-lithiation and (**f**) cycled-HF-MXene. Ti atoms, C atoms, and surface terminations are marked by blue dots, yellow dots, and red dots, respectively. **g** The EELS mapping of Ti, C, O, and Li of cycled-HF-MXene sample. Black represents the lowest intensity. **h** A schematic showing the solvents co-intercalation in HF-$Ti_3C_2T_x$.

for MS-MXene which results in the evolution of small amount of $CO_2$ (m/z = 44) in the same range. A similar trend is also observed for the EC component (m/z = 43) where only the HF-MXene exhibits a characteristic peak after vacuum drying (Supplementary Fig. 14). Thus, the TPD-MS confirms the presence of co-intercalation of solvents only in HF-MXene while lithium ions are entering desolvated in the interlayer of MS-MXene before the SEI layer forms. Considering the results of in-situ-XRD tests in this work and the previous work[47], this O-enriched surface is conducive to attracting Li ions and facilitating Li ions stripping from the solvent shells before the SEI is formed to block the solvents[26,27].

EDS mapping of the cross sections of MS-initial-lithiation, cycled-MS-MXene, HF-initial-lithiation, and cycled-HF-MXene samples was conducted to study the distributions of F and O elements at the MXene/electrolyte interface (Fig. 4e, f and Supplementary Fig. 15) following the electrochemical lithiation. A thin layer of platinum (Pt) was sputtered onto the MXene surface to protect the SEI layer formed during the first reduction cycle, before any experimental procedures. Based on the F signal in Fig. 4e, a roughly 200 nm-thick SEI layer is observed on the surface of MS-$Ti_3C_2T_x$. This SEI layer contains O atoms, likely resulting from the decomposition of carbonate solvents, and also comprises a significant amount of fluorine, known to enhance the stability of

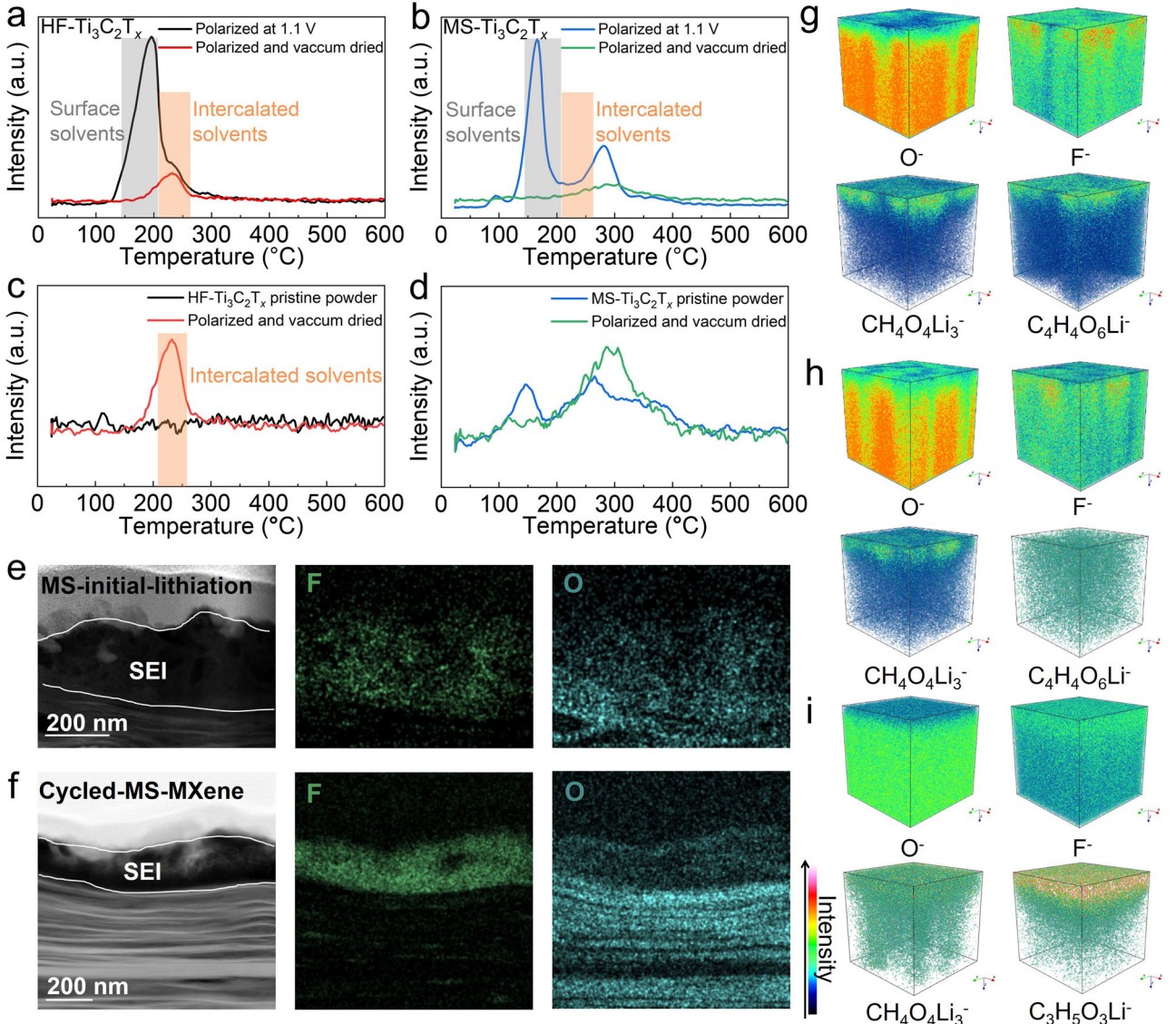

**Fig. 4 | Characterizations of solvents and SEI layers.** TPD-MS measurement of DMC component (m/z = 45) of (**a**) HF-$Ti_3C_2$ electrodes polarized at 1.1 V and further vacuum dried for 80°C, (**b**) MS-$Ti_3C_2$ electrodes polarized at 1.1 V and further vacuum dried for 80°C, (**c**) expanded graph of vacuum dried HF electrodes and pristine HF powder, and (**d**) expanded graph of vacuum dried MS electrodes and pristine MS powder. EDS mapping of the cross section of (**e**) MS-initial-lithiation electrode materials[48]. The stability of the SEI layer on MS-MXene after cycling is confirmed by Fig. 4f, which shows almost no change in the thickness with the initial SEI layer formed at the first cycle. In contrast, a thin, ~30 nm-thick, SEI layer is formed on the surface of the HF-MXene sample after the first lithiation cycle (Supplementary Fig. 15a), as can be seen from the continued presence of F and O signals, confirming recent results we obtained by NMR spectroscopy[49]. Importantly, the SEI layer of the HF-MXene is no longer visible after three lithiation/delithiation cycles (Supplementary Fig. 15b).

and (**f**) cycled-MS-MXene illustrating the cycling stability of the SEI layers. 3D reconstructions of different components in SEI layers via TOF-SIMS of (**g**) the initial cycle via CV test of MS-$Ti_3C_2T_x$, (**h**) the third cycle via CV test of MS-$Ti_3C_2T_x$, and (**i**) the initial cycle via CV test of HF-$Ti_3C_2T_x$. Source data are provided as a Source Data file.

High resolution transmission electron microscope (HRTEM)[50], XPS[48], time of flight secondary ion mass spectrometry (TOF-SIMS)[51], and Fourier transform infrared (FTIR)[52] techniques were used to further analyze the SEI layers (see Fig. 4g–i and Supplementary Figs. 16–21). TOF-SIMS is capable of visualizing the distributions of different components from the SEI/electrolyte interface (top) to the SEI/MXene interface (bottom) (Fig. 4g–i and Supplementary Figs. 18 and 19). The SEI formed on MS-MXene is mainly composed of

O-containing inorganic ($Li_2CO_3$) components generated by the reaction between O-terminations and carbonate solvents, and the organic components in the outer layer degrade after cycling. Differently, the inorganic phase of the SEI layer on HF-MXene contains mainly F-based products resulting from the reaction between F-terminations and electrolytes with more organic components in the outer layer, in agreement with our previous results on the SEI layer formation[49]. Combined with other characterization techniques (Supplementary Figs. 18–21), the analysis of the SEI layer shows that a ~200 nm-thick dense and stable SEI layer is formed on the surface of MS-MXene, enriched with inorganic components. The presence of such a thick and robust SEI layer, enriched with inorganic components with high conductivity, could facilitate the desolvation of the Li ions and their transfer through the electrochemical interface. The Li ion desolvation results in improved charge transfer, increased capacity, and long-term stability of the electrodes by preventing the exfoliation of MXene interlayers[49,53,54]. It is similar to what was observed in graphite negative

electrodes of Li-ion batteries[54]. Differently, the SEI layer formed on HF-$Ti_3C_2T_x$ during the first lithiation cycle is about ten times thinner and contains an important part of organic compounds. Combined with the presence of inorganic LiF, this could explain the instability of the SEI layer, which is found to disappear after 3 cycles of Li intercalation/deintercalation, offering access of solvated Li ions to the interlayer space of HF-MXene[49].

Molecular dynamics (MD) simulations were conducted to elucidate the distinct electrochemical behaviors observed in HF-MXene and MS-MXene samples (see Supplementary Figs. 22 and 23; Supplementary Videos 1 and 2). Simulations reveal that only desolvated Li ions can intercalate into the interlayers of negatively charged MS-MXene, whereas solvent molecules intercalate together with Li ions into HF-MXene. Although the simulations cannot account for the presence of a SEI layer on the MXene outer surface, these findings are consistent with our experimental observation. Then, $CH_3$ groups from DMC solvent were found to be in close interaction with F-terminations on the HF-MXene surface. MD results also reveal that the distance between intercalated Li ions and -O terminations is about 0.3 Å closer in MS-MXene compared to HF-MXene (Supplementary Fig. 23c). Together with the formation of a stable SEI layer onto MS-MXene, the high affinity of the $CH_3$ groups from the electrolyte solvent for the F-termination groups of the HF-MXene well support the difference in the electrochemical behaviors between the two MXenes.

To summarize, by coupling electrochemical characterizations, direct observation of the atomic structures of MXenes and NMR and TPD-MS measurements, we evidence the atomic-scale desolvation and efficient intercalation of desolvated Li ions in O- and Cl-terminated MXene. The capacity enhancement mechanism has been clarified. O-rich surface chemistry of MS-MXene forms a thick (~200 nm) and dense SEI layer, facilitating efficient Li ion desolvation. Furthermore, the reinforced attraction between desolvated Li ions and O-rich surface functionalities of MS-MXene interlayers improves the charge storage capacity, which is in agreement with modeling studies[26,27]. Our understanding of the effect of ion desolvation on pseudocapacitive energy storage sheds light on studying the ionic nature in the bulk of materials (or at the polarized interface) and tailoring the nature of electrodes, promoting the development of fast and sufficient energy storage systems. These findings establish the construction principle for a broad family of materials with tailored surface functionalities for high-performance energy storage applications.

## Methods

### Preparation of pristine MS-MXene and HF-MXene

One gram $Ti_3AlC_2$ MAX powder (400 mesh, 11 Technology Co., Ltd) was mixed with 0.6 g KCl (AR, Chengdu Chron Chemical Co., Ltd), 0.76 g LiCl (AR, Chengdu Chron Chemical Co., Ltd) and 2.1 g $CuCl_2$ (AR, Chengdu Chron Chemical Co., Ltd). After hand-grinding of the mixture in a mortar for 15 minutes, the mixture was heated to 750 °C with argon flow. The obtained powders were washed with deionized water and 0.1 M APS solution (AR, Chengdu Chron Chemical Co., Ltd) to remove the impurities. The MS-MXene powders were then obtained after drying in vacuum at 80 °C for 36 h, resulting in a decrease in the Cl content.

The mixture of 1 g $Ti_3AlC_2$ (400 mesh, 11 Technology Co., Ltd), 1 g LiF (AR, Macklin), and 20 mL of 9 M HCl solution (36–38 wt%, Sinopharm Chemical Reagent Co., Ltd) was stirred at 35 °C for 1 day. After washing the mixture with deionized water until the pH reached ~7, the sediment was dispersed in deionized water and sonicated in an ice bath for 30 minutes. After centrifuging the solution at 3500 rpm for 1 h, the HF-$Ti_3C_2$ solution was collected.

### Preparation of pristine and lithiated samples for electrochemical tests and characterizations

The HF-MXene electrode was prepared by vacuum filtration through the Celgard 3501 polypropylene membrane followed by drying. The MS-MXene electrode was made by mixing MS-MXene powder, polytetrafluoroethylene (60 wt%, Aladdin), and conductive carbon black (XFNANO) with the weight ratio of 78:7:15 in a mortar at room temperature followed by rolling into plates and drying in vacuum. Then the electrodes were cut into plates with the diameter of 14 mm using a stopper borer. The MS- and HF-MXenes electrodes for electrochemical tests shared the same mass loading of active material of ~0.8 mg cm$^{-2}$.

The lithiated samples for iDPC-STEM, HAADF-STEM, and SEI characterizations were polarized from OCV (about 2.7 V versus Li$^+$/Li) down to 0.2 V (vs. Li$^+$/Li) at the scan rate of 0.5 mV s$^{-1}$ (see details in electrochemical tests section below). Samples were kept at 0.2 V (vs. Li$^+$/Li) until a steady state current was reach. Then, the coin cells were disassembled in a glove box and the lithiated MXene electrodes were immersed in the DMC (99.5%, Sinopharm Chemical Reagent Co., Ltd) with gentle shaking for a few seconds to remove the LP30 (1 M LiPF$_6$ in 1:1 vol/vol ethylene carbonate/dimethyl carbonate, no additives, AR, Sigma-Aldrich) in the surface followed by drying overnight in vacuum. The samples were stored in a glove box and kept in vacuum before characterizations.

### Materials characterizations

The XRD characterizations of the MXene samples were conducted on an X-ray diffractometer Bruker AXS-D8 (Cu/Ka radiation). The iDPC and HAADF images were conducted in spherical-aberration-corrected TEM instruments (Spectra 300 X-CFEG (FEI) operated at 300 kV, and also ARM200F (JEOL) operated at 200 kV for comparison). The beam currents were 0.05 nA and 0.01 nA. The convergence semi-angle was 25 mard. The collection angle was 49–200 mard and the dwell time was 2 µs; The aberration coefficients were measured as: $A_1 = 2.54$ nm; $A_2 = 26$ nm; $B_2 = 19$ nm; $C_1 = 1.58$ nm; $C_3 = 469.8$ nm. EELS spectra were acquired on the Gatan 965 GIF QUANTUM ER with the GIF Continuum K3 camera at 200 kV. The energy dispersion of EELS-STEM is 0.45 eV Ch$^{-1}$. The samples for STEM imaging were prepared by a Thermo Scientific Scios 2 DualBeam FIB system. 30 kV and 0.5–0.03 nA were applied to cut the samples into the slices. Five kilovolts and 48 pA were applied to thin the samples finely by FIB cutting. The samples were further cleaned by 2 kV and 43 pA.

TPD-MS measurements on pristine MS- and HF-MXene samples, lithiated MS- and HF-MXene samples were performed under an inert atmosphere (Ar, 100 mL min$^{-1}$). The sample (10–20 mg) was placed in a thermo-balance and heated up to 600 °C (800 °C for pristine HF-MXene) at a rate of 10 °C min$^{-1}$. The decomposition products (gas evolved) were monitored by online mass spectrometry (Skimmer, Netzsch, Germany). The m/z was chosen where m/z 43 for EC and m/z 45 for DMC because (even if they do not have the maximum relative intensity) they are the ones that are independent for each moiety and have high enough relative intensity.

HRTEM and EDS characterizations of SEI layers were performed with an FEI Titan G 2 80–200 ChemiSTEM. EDS characterizations of Ga were performed with a Field Emission aberration-corrected Transmission Electron Microscope Hitachi HF5000 at 200 kV. XPS depth profiles were obtained via argon ion etching with a depth of 1 µm of pristine MS- and HF-MXene on the Thermo Scientific ESCALAB 250Xi. The etching depths are 200 nm and 30 nm for lithiated MS-MXene and lithiated HF-MXene, respectively to analyze the components of SEI layers. FTIR was conducted with a Bruker Vertex 70. TOF-SIMS was measured with a PHI NanoTOF II.

$^7$Li ssNMR measurements were conducted on Bruker 600 MHz spectrometer in the same condition. The electrodes with the same mass were polarized to 0.2 V (vs. Li$^+$/Li) during the initial and the third CV cycles at 0.5 mV s$^{-1}$. Four lithiated samples were immersed in the DMC with gentle shaking for a few seconds to remove the LP30 in the surface followed by drying overnight in vacuum. The samples were kept in vacuum before conducting the tests.

All the characterization results are reproducible, and each measurement was repeated at least three times.

## In-situ-XRD

In-situ-XRD tests were performed on a Bruker Advance D8 using a two-electrode electrochemical cell (not custom-built) with LP30 as the electrolyte, MS- and HF-MXene as working electrodes, and the lithium metal as the counter electrode. The evolution of the (002) peak between 5° and 9° was monitored during the GCD tests at the current density of $0.03 \, A \, g^{-1}$. The in-situ-XRD test results are reproducible, and we made three cells for each MXene.

## Molecular dynamic simulations

Supplementary Fig. 22 shows the molecular simulation system of MS-MXene and HF-MXene, immersed into 1 M LiPF$_6$ in EC/DMC electrolyte. The stoichiometric ratios of the terminations on the MS-MXene and HF-MXene surfaces were estimated based on experimental data. The distance between MS-MXene and HF-MXene layers were set according to the experimental XRD results. The Large-scale Atomic/Molecular Massively Parallel Simulator classical MDs code package was utilized for all simulations[55]. The schematic atomic models were visualized and rendered using the OVITO software[56]. The bonding and non-bonding interactions among the atoms were characterized utilizing the ClayFF force field[57]. The force field parameters, encompassing Lennard-Jones potential, point partial charges, harmonic bonds, and harmonic angles, were adopted from previous studies[20,58,59]. The particle-particle particle-mesh (PPPM) method was used in the k-space to handle long-range electrostatic interactions. The simulations were conducted in the Canonical ensemble (NVT) at a target temperature of 350 K, using an integration time step of 1 fs. To investigate the dynamic process of Li ions and solvents embedded between the MS-MXene and HF-MXene layers in the electrolyte upon charging, −2 V potentials were applied to the MS-MXene and HF-MXene electrodes using the Constant Potential Method (CPM), respectively[60]. This CPM method maintains a constant potential difference between the electrodes and captures charge fluctuations on the electrode atoms and the ions dynamic of electrolytes throughout the simulation. The calculated radial distribution functions of MXene surface terminations (F and O) with CH$_3$ from intercalated solvents in HF-MXene are shown in Supplementary Fig. 23a, b. The calculated radial distribution functions of intercalated Li ions with O terminations of MS-MXene and HF-MXene are displayed in Supplementary Fig. 23c.

## Electrochemical tests and calculations

The electrochemical tests were performed on an electrochemical workstation (PGSTAT302N, Metrohm Autolab B.V.) using two-electrode coin cells fabricated in the glovebox (O$_2$ < 0.1 ppm and H$_2$O < 0.1 ppm). Lithium metal was used as the counter electrode (99.95%, the diameter of 12 mm, Guangdong Canrd New Energy Technology Co., Ltd) with the thickness of 15.6 mm, and the Whatman GF/A glass microfiber filter (the thickness of 260 μm, the diameter of 19 mm) was used as the separator. The CVs were tested at the scan rates of 0.1 to 100 mV s$^{-1}$, and GCD tests were cycled at the current density of 0.1–5 A g$^{-1}$. All the electrochemical results are reproducible, and we made three cells for each measurement.

The capacities C (C g$^{-1}$) of MS- and HF-MXene electrodes were calculated from the anodic scan of each CV curve using the Eq. 1:

$$C = \frac{\int i \, dt}{m} \quad (1)$$

The $b$ value was calculated using the Eq. 2:

$$i = av^b \quad (2)$$

where $i$ is the anodic peak current, $v$ is the scan rate, $a$ and $b$ are parameters. The $b$ value reflects the charge storage mechanism of electrodes. $b = 0.5$ stands for a semi-infinite diffusion-controlled charge storage process, while $b = 1$ means a surface-controlled process[8].

## Data availability

All data that support the findings of this study are presented in the manuscript and Supplementary Information, or are available from the corresponding author upon request. Source data are provided with this paper: https://doi.org/10.6084/m9.figshare.28532912.

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

## Acknowledgements

Z.B. was funded by the National Natural Science Foundation of China (No. 52076188), National Key Research and Development Program of China (No. 2023YFB2405800), and Key R&D Program of Zhejiang (No. 2023C01128). K.X. was funded by the National Natural Science Foundation of China (No. 22202102). Z.L. thanks the National Natural Science Foundation of China (No. 52472228). H.S. was funded by the National Natural Science Foundation of China (No. 22309202) and Gusu Leading Talents Program (ZXL2023190). Q.Y. was funded by the National Key Research and Development Program of China (No. 2023YFB2405800),

National Natural Science Foundation of China (No. 52325102), and Key R&D Program of Zhejiang (No. 2023C01128). P.S. and P.-L.T. are grateful for support from the European Research Council (ERC Synergy Grant MoMa-Stor #951513), the LABEX STOREX, and the PEPR Batteries ANR-22-PEBA-0003.

## Author contributions

Z.B., Q.Y., J.Y., and P.S. conceived the project. Z.B., R.W., and Z.L. discussed the conceptualization of the work and experimental designs with Q.Y., J.Y., and P.S. Z.B., R.W., B.W., and S.S. fabricated materials and performed experiments. Z.B., R.W., Y.Z., Y.S., and Q.Y. performed HAADF-STEM, iDPC-STEM, and EELS-STEM characterizations. Z.B., R.W., J.Y., and K.C. performed ssNMR tests. Z.B., S.S., K.G., E.R.-P., and P.S. performed TPD-MS measurements. K.X., H.S., and P.S. conducted MD simulations. Z.B., R.W., S.S., K.G., Z.L., H.S., Q.Y., J.Y., K.C. P.-L. T. and P.S. analyzed the data and interpreted the results. All the authors contributed to the writing of the manuscript.

## Competing interests

The authors declare no competing interests.
