## [Transparent Peer Review file · Nature Communications]

Ion desolvation for boosting the charge storage performance in Ti₃C₂ MXene electrode

Corresponding Author: Professor Patrice Simon

Version 0:

Reviewer comments:

Reviewer #1

(Remarks to the Author)

This manuscript demonstrated the intercalation of Li ions from a non-aqueous electrolyte in two-dimensional metal carbides at the atomic scale. However, this reviewer feels that the data presented are not convincingly enough to justify their correctness; therefore, this manuscript is not recommended for publication in Nature Communication. Additional comments are as follows:

1. MXenes are 2D materials so that I assume most of the layered materials lay down on the TEM grids. How do the authors observe the cross-section of the layered structures? If it was done by randomly selecting the luckily oriented samples, then are the images shown here representative? Moreover, low magnification morphology should be provided other than the local views in Figs 1 and 3.
2. Although contrast can be observed in the interlayer and elements can be detected by EELS, how do the author obtain one-to-one correspondence for proposed atomic model and the image. For example, the weak interlayered contrast in the middle of Fig 3b was assigned to be Li atoms. Why could not be oxygen atoms, or at least partially oxygen atoms? Besides, the signals of O and Li in EELS are almost in the level of noise. So I could not agree with the authors strong statements of direct atomic-scale observation of full desolvated Li ions or the embedded "solvent" layer.
3. The authors mentioned that "The stability of the SEI layer on MS-MXene after cycling is confirmed by Fig. 4e, which shows almost no change with the initial SEI layer formed at the first cycle." It is difficult to observe that there was no change in the morphology of the SEI before and after cycling. Before cycling, the SEI surface has shallow contrast particles, while after cycling, the SEI surface forms a shallow contrast film with a thickness of approximately 100 nm. In addition, the SEI image before cycling should be included in main text for better comparison.
4. Fig. 2b showed the b values were inconsistent with the descriptions in the text "the b value obtained from Equation (see methods) for MS-Ti₃C₂T_x is larger than that of HF-Ti₃C₂T_x (of 0.91 vs 0.71, respectively) in the range of 0.5 to 100 mV s⁻¹".

Reviewer #2

(Remarks to the Author)

This manuscript systematically investigates the ion desolvation and solvent-ions co-intercalation behaviors of HF-Ti₃C₂ and MS-Ti₃C₂ MXene with different surface chemistries during lithium-ion intercalation by utilizing advanced characterization tools such as IDPC-STEM, ssNMR, and TPD-MS. It is found that the O- and Cl-terminated MS-MXene can achieve complete desolvation intercalation of Li ions and form a stable SEI layer, resulting in increased capacity and power performance. This research is of great significance for understanding the role of ion desolvation in energy storage. However, several critical issues should be addressed before the manuscript can be considered for publication in Nature Communications:

1. This manuscript proposes that O- and Cl-terminated MS-MXene allows complete desolvation intercalation of Li ions, whereas the F-, OH-, and O-terminated HF-MXene is accompanied by solvent co-intercalation. While the specific surface functional groups and their interactions with the lithium ions and solvents are undoubtedly important factors, other structural and compositional characteristics of the MXenes may also play a significant role. Are such differences simply due to different in surface chemistry, or do other factors also play a role?
2. When preparing HF- and MS-MXene samples for ⁷Li ssNMR spectroscopy experiments, how to control the washing away of the surface electrolyte without affecting the solvents between interlayers in the MXene?
3. It is recommended to provide higher resolution EELS-STEM mapping and EDS mapping of SEI layers.

4. Whether different electrolytes interact with the -O or -F terminations on the surface of MXene and ultimately affect the formation of SEI and charge storage capacity in this work. The following literature can be consulted (Energy Environ. Sci., 2024,17, 3021-3031).
5. Although the experimental data and characterization results support the conclusions, and the analysis and discussion are relatively comprehensive. However, there are some contents that can be further explored, such as the formation mechanism of SEI layer and the specific effect of the desolvation process on the electrochemical performance.
6. Figures need to be further standardized.
7. The format of references is not uniform, e.g., the journal name is the full title in ref.38.

Reviewer #3

(Remarks to the Author)

The paper provides an in-depth investigation into the impact of ion desolvation on the charge storage performance of Ti₃C₂ MXene materials with varying surface chemistries, using a combination of experimental characterizations and molecular dynamics simulations. The authors successfully elucidate the relationship between ion-electrode interactions and surface terminations, demonstrating how desolvation can significantly enhance charge storage capacity. The study offers a comprehensive understanding of how tailoring surface functionalities can lead to the development of high-performance energy storage devices.

It highlights the advantages of O- and Cl-terminated MXenes in achieving complete ion desolvation and stable SEI formation, which contribute to enhanced electrochemical performance. The findings emphasize the critical role of surface chemistry in influencing ion transport and charge storage mechanisms. Overall, this work serves as a valuable resource for researchers in the field, providing new insights into the design principles of MXene-based energy storage materials. Before publication, the reviewer kindly recommends that there are several points needed to be addressed in revised manuscript.

- 1) In the Introduction section, it discusses the importance of understanding electrolyte ion dynamics and differentiates between capacitive (non-Faradaic) and battery-like (Faradaic) processes. However, the transition between these two processes is not clearly explained, and the reader may find it difficult to understand the relationship between ion desolvation and the resulting pseudocapacitive behavior. It is recommended to include a brief explanation or diagram to illustrate how ion desolvation influences the charge storage mechanism in MXenes. There should be a clarification of the charge storage mechanisms in the introduction.
- 2) In Figure 1 (a-c), it is represented the same materials (MS- Ti₃C₂ Tx with the d-spacing of 1.1 nm), but it seems like Ti atoms which is presented as blue dots are marked differently. If so, there should be a specific explanation why there is a difference between the atomic structures for same materials.
- 3) In Figure 1 (e), the intensity of the C and O elements in the STEM-EELS image is too weak to clearly distinguish these elements. To enhance the clarity and provide more evidence, it is recommended to include additional supplementary information regarding the presence and distribution of these elements.
- 4) The author compares HF- Ti₃C₂ and MS- Ti₃C₂, emphasizing the role of different surface terminations. However, it would be beneficial to provide more justification for why these specific MXene surface chemistries were selected. Expanding on the rationale for using F-, OH-, O- (HF- Ti₃C₂) and O-, Cl- (MS- Ti₃C₂) terminations, and their expected impact on ion desolvation and intercalation, would strengthen the paper. There should be a justification for the selection of MXene surface chemistries.
- 5) In the Methods section, Although the experimental methods are described, some details are missing, such as the specific conditions for the ion intercalation experiments and the justification for choosing particular characterization techniques like IDPC-STEM. Providing additional information on these points would improve the reproducibility and understanding of the experimental procedures.
- 6) The paper reports the formation of a solid electrolyte interface (SEI) layer in MS-Ti₃C₂, contributing to enhanced charge storage. It would be valuable to include a more detailed discussion on how the SEI layer's composition and thickness were determined and how these factors influence the long-term stability and performance of the MXene electrodes.
- 7) The author mentions using molecular dynamics (MD) simulations to support the experimental findings. It would be helpful to provide more information on how these simulations were set up, including the specific models and parameters used. Additionally, discussing any limitations of the MD simulations and how they complement the experimental results would strengthen the discussion.
- 8) There are some typo errors in this paper below.
 - 8-1) For the consistency throughout the paper, the term "Li+ ions" should be replaced to term "Li ions"
 - 8-2) In Introduction section, it is needed to develop a consistency in terminology. The terms "pseudocapacitive" and "capacitive" are used to describe the charge storage mechanisms. For consistency and clarity, it is suggested to define these terms clearly in the Introduction and ensure their consistent use throughout the text.

Version 1:

Reviewer comments:

Reviewer #1

(Remarks to the Author)

In the revised Figure 1g, the authors provide elemental detection using EELS. I can see an obvious mistake in this data. For oxygen element, the major edge is the K edge (~530 eV), while the minor edge is the L edge (~20 eV). It is contradictory that

in the figure where signal from minor edge can be observed clearly whereas the major K edge signal can not be seen. It is doubtful whether the 20 eV peak really originates from oxygen. The authors should check their data carefully. Even if I assume the 20 eV signal is from oxygen. It is obvious the the concentration of oxygen is much higher than lithium from many of the EELS spectra provided by the authors. The authors did not response to my comment2 that how can it be excluded that the interlayer contrast is not oxygen, but solely lithium. The strong conclusion is not reliable. Moreover, I can also provide other explanations for such contrast. For example, as can be seen from low magnification images that the layer is heavily wavy. The authors also claim that they chose specific postions where the samples are in zone axis and clear images can be taken. Then I would propose that the interlay contrast could be originated from the wavy natuer of the layer, so that inclined portions can be existed in the pathway of the electron beam and therefore the 2D projection shows artifacts. In all, the improvements made by the authors still could not support their strong conclusion.

Reviewer #2

(Remarks to the Author)

The authors well considered my comments, now it is acceptable for publication now.

Reviewer #3

(Remarks to the Author)

The authors have provided detailed responses and made necessary modifications to strengthen the overall quality and clarity of the paper. The authors' efforts in improving the manuscript are commendable, and the current version is well-prepared for dissemination. As a result, the authors have thoroughly addressed all the concerns raised by the reviewer in the revised manuscript, and it is now recommended for publication.

Version 2:

Reviewer comments:

Reviewer #1

(Remarks to the Author)

In the last round of review, I did not see O-K signal in the middle of Fig1.g, as compared with Ti signal. However, in this updated document, the data has been changed. The O-K signal is comparable to Ti signal. Besides, the authors claimed that they have mistakenly used a wrong image in the manuscript in the last round of review. From these aspects, I'm quite confused with the data which is inconsistent and it seems the authors are not rigorous, which does not meet the criterion of Nature journals.

For the identification of Li ions, in my personal view, the conclusion drawn from IDPC-STEM image contrast and weak EELS signal is far from enough. Whereas the interlayered Li is a critical point for this manuscript, therefore I could not be positive.

Reviewer #4

(Remarks to the Author)

First, I read the entire paper and found the results to be of high quality and significant importance. The characterizations presented (iDPC, EELS maps) are essential for supporting the main conclusions of the authors. I also recognize the high quality of the electron microscopy (EM) experiments on these extremely challenging samples. Therefore, the results should be evaluated carefully.

However, I believe there are too many experimental unknowns for the paper to be ready for publication in its current form. Despite focusing on the EELS evaluation as requested, I also reviewed the iDPC images, as I am somewhat familiar with the technique, and the images are closely linked to the EELS maps/spectra.

Key Issues and Questions:

1. Beam Sensitivity of MXenes:

- o MXenes (including Ti₃C₂T_x) are extremely beam-sensitive, but this is not addressed in the paper. This sensitivity could impact both the iDPC and EELS maps, especially at 300 kV.
- o Experimental details, such as the beam current, are missing. For instance, was DCFI used? Typically, dDPC is preferred—did the authors examine these images?
- o What is the EELS energy dispersion? Was Fourier-log deconvolution applied?

2. Sample Preparation:

- o Sample preparation is critical for obtaining representative EM results, yet minimal information is provided. For example, if a FIB was used with Ga ions, these could interact with the MXenes. Was Ga detected in the EDX maps? Was low temperature used to minimize beam damage?

3. EELS Spectra:

- o The EELS spectra in the supplementary information are insufficiently detailed: i. The 50-70 eV range is not only the location of the Li K edge but also the Ti M_{2,3} edge. Double plasmon effects might also be present. A broader energy range should be shown to assess the influence of the Ti edge or confirm if the bumps are just noise. ii. The authors do not clarify whether the spectra represent single pixels, pixel sums, or specific areas. iii. How were the maps produced? How was the background in the Li-K region removed? iv. What does the spectrum at the L_{2,3} edge of Ti in the gaps (Figure 3g) look like? The maps do not convincingly show the absence of Ti, and the contrast scale could obscure its presence.

4. Resolution of EELS Spectra:

o The resolution (or zoom) of the EELS spectra is inadequate to confirm correspondence with typical MXene spectra (e.g., Figure 1g). The carbon K edge, in particular, does not resemble commonly reported results. Comparison with the literature is necessary.

o To prove the presence of solvents in the interlayer gap, why did the authors not present the carbon K edge in this region? The fine structures of the carbon K edge differ significantly between carbonates (solvents) and MXenes.

5. iDPC Images:

o For Figure 3e: i. While iDPC interpretation is linked to atomic number (Z), it is highly sensitive to orientation and order. This sensitivity exceeds that of HAADF due to the composite processing of quadrant-detector signals. Thus, lower contrast in the interlayer gap does not necessarily indicate lighter elements. ii. Within the Ti_3C_2 slab, stripes and double spots in various areas suggest that the contrast is complex and prone to misinterpretation in disordered samples (e.g., Figure 1a). iii. The contrast scales for iDPC and HAADF (Figure 3e) should be consistent to determine if there is intensity in the gap in HAADF (it appears so). If true, this could indicate the presence of additional heavy atoms (in limited proportions) in the gap. iv. Were iDPC and HAADF images acquired simultaneously or sequentially? If sequentially, in what order?

6. Ordering of Solvent Molecules (Figure 3f and h):

o The periodic contrast observed is unlikely to be due to Li ions (invisible under these conditions). It is more plausibly attributed to carbonate molecules.

o At room temperature, solvent molecules likely spin rapidly, making their ordering improbable. Under electron beam exposure, these molecules typically undergo radiolysis. Are the images repeatable? Do they depend on dose? The electron-conducting properties of MXenes might aid imaging of these sensitive molecules, but this needs to be verified and discussed.

Final Comments:

These questions are highly technical, but I believe the electron microscopy experiments are crucial for this publication. The standards of Nature Communications are exceptionally high, and the authors must provide sufficient information for other researchers to reproduce the experiments. Currently, I do not think there is enough detail to ensure reproducibility.

Version 3:

Reviewer comments:

Reviewer #4

(Remarks to the Author)

The authors have done a commendable job in complementing their manuscript, and I would like to thank them for their efforts.

I still have a few comments (see below). However, in my opinion, they provide sufficiently strong evidence of their remarkable findings, along with enough details for others to reproduce them. Overall, their work meets the standards expected for publication in Nature Communications.

- On beam damage and associated comments: The explanations are now much clearer and more convincing—thank you.
- On fine structures in EELS spectra: The dispersion of 0.45 eV/pixel accounts for the lack of clarity and the absence of subtle information in high-energy edges. The authors might consider using the DualEELS option in future studies (e.g., with 0.1 eV dispersion for two energy ranges) to gain deeper insights into chemical bonding, particularly in the interstitial gap. However, this goes beyond the scope of the present study.
- On the presence of π and σ peaks in MXene layers: I would reconsider the authors' addition suggesting the presence of π^* and σ^* peaks. Since carbon atoms are in an octahedral environment, π^* bands do not exist. The presence of two peaks and their evolution can certainly be discussed, but without assigning them a specific nature, as the interpretation is more complex than simply π^* (see studies on DFT band structures of $Ti_3C_2X_n$ compounds).
- On EELS mapping (Comment 3iii): Typically, to produce EELS maps, the intensity in the relevant region (e.g., the Li K-edge) is extracted after background removal. I do not fully understand the response to my previous comment on this matter, but Figure R5 is quite informative. Since it has now been included as Supplementary Figure 13, this resolves my concern.
- On iDPC imaging: Thank you for the clarification. I particularly appreciate the response to Comment 5, where the authors adopt a more cautious approach and thoroughly discuss their conclusions based on these images. I agree that their overall experimental findings strongly support the presence of Li ions in the interlayer gap—this is particularly well supported by Supplementary Figure 13. Regarding the iDPC images, I believe it would be appropriate to use the term “plausible” in the manuscript. Since a titanium signal is detected in the interlayer region (Supplementary Figure 13), these atoms may also appear in the iDPC images of this area. I also appreciate the authors making their image database available to readers.
- On Comment 6 (ordering of molecules): While this is not the main result and is therefore not critical for acceptance, a non-specialist in STEM imaging techniques might misinterpret the findings as proof of molecular ordering, which has not been demonstrated (as acknowledged by the authors). It would be helpful to clarify this point further.

Conclusion:

The authors' responses have largely addressed my concerns. Aside from minor revisions regarding the interpretation of π^* - σ^* features and the iDPC imaging discussion (where I trust the authors to make the necessary adjustments), I see no reason why this work should not be published in Nature Communications.

Response to Referee Reports

Authors would like to thank all the Referees for their critical comments and thoughtful suggestions. As you will see below, we conducted additional IDPC-STEM and EELS-STEM characterizations, modified the Figures and the statements accordingly to better highlight the reliability and quality of our manuscript. New STEM images were acquired following Referee's comments, and were moved to the main text to replace the original ones. To ensure clarity, we have retained the modified Figures in the responses provided to the Referees, which accounts for the document's length of 39 pages. All the changes in the text were marked in blue and the changes in Figures are marked in blue and bold in the label of the Figures.

For the convenience of your reading, we marked the page number for the response to each Reviewer here:

Response to Reviewer #1 is from page 1 to 17.

Response to Reviewer #2 is from page 18 to page 27.

Response to Reviewer #3 is from page 28 to page 39.

Reviewer #1 (Remarks to the Author)

This manuscript demonstrated the intercalation of Li ions from a non-aqueous electrolyte in two-dimensional metal carbides at the atomic scale. However, this reviewer feels that the data presented are not convincingly enough to justify their correctness; therefore, this manuscript is not recommended for publication in Nature Communication.

General response: We would like to thank Referee#1 for their constructive comments and criticisms that helped us to improve the quality of the manuscript. The Referee will see that, following their recommendations, additional characterizations have been achieved to provide detailed, point-by-point answers to each of the concerns raised. All the changes made to the revised manuscript have been highlighted in the main text. These important suggestions help us better highlight the reliability of the results of our STEM characterizations. We believe that all the concerns have been addressed and hope that the Referee will find this revised version suitable for publication in Nature Communications.

Specific Comment 1: MXenes are 2D materials so that I assume most of the layered materials lay down on the TEM grids. How do the authors observe the cross-section of the layered structures? If it was done by randomly selecting the luckily oriented samples, then are the images shown here representative? Moreover, low magnification morphology should be provided other than the local views in Figs1 and 3.

Response:

We appreciate your insightful comments. To visualize the layered MXene structure, we used focused ion beam (FIB) milling to obtain cross-sectional samples of the MXenes, as illustrated in additional Figure R1 below, which is different from the direct observation of the particles in conventional HRTEM technique. Taking the MS-MXene as example, we randomly choose particles larger than 10 μm in size, suitable for FIB cutting, and cut out the cross-section slices. Then the slices were transferred and adhered to a half-circle TEM grid using platinum plating, then further thinned to tens of nanometers by ion milling for imaging.

In response to the Referee's suggestion, we conducted additional HAADF-STEM and IDPC-STEM characterizations and acquired low magnification images, see examples in Figures R2, R3 and R4 below, that fully confirm that the images are representative of the structure of the material. We then

replaced all local STEM images of the main text with these new larger region images. More specifically, Figure R2 was added as Figure 1a and Supplementary Figure 8, Figure R3 was included in Figure 3 and local STEM images of pristine MXenes in Figure 1 were replaced with Figure R4 in the revised version of the manuscript. Other low magnification images were added as Figure 1b and Supplementary Figure 8 (see below). Finally, additional HAADF-STEM and IDPC-STEM images at lower magnifications of the four lithiated samples have been collected and added to Supplementary Figures 9 and 11 (see below).

In all these images, the observed phenomena remained consistent across these observations; the images are then representative of the samples. We have revised the manuscript to include more details on TEM sample preparation and noted that we imaged various local regions across four different groups of samples. However, one has to keep in mind that obtaining a comprehensive view of the atomic structure remains challenging due to local bending and variations in thickness, which may affect the focus and contrast of the HAADF and IDPC-STEM images.

Fig. R1: The preparation of MS-MXene FIB milling samples. The particle cut by FIB is marked by a white circle (left). The right image shows the FIB milling slice with tens of nanometers in thickness.

Fig. R2: The HAADF-STEM images of the cross-section of the pristine and lithiated MS-MXene FIB slices at low magnification.

magnification. This Figure was added in Figure 1a and Supplementary Figure 8 in the revised manuscript.

Fig. R3: HAADF-STEM images and the corresponding IDPC-STEM images of larger regions of MS-initial-lithiation, cycled-MS-MXene, HF-initial-lithiation and cycled-HF-MXene samples. Ti atoms, C atoms, surface terminations and Li atoms are marked by blue, yellow, red and purple dots, respectively. **These Figures were added in Figure 3 in the revised manuscript.**

Fig. R4: HAADF-STEM images and the corresponding IDPC-STEM images of larger regions of pristine MS- and HF-MXene samples. Ti atoms, C atoms and surface terminations are marked by blue, yellow and red dots, respectively. **These Figures were added in Figure 1 in the revised manuscript.**

Changes to the manuscript:

1. *Figures 1, 3 and Supplementary Figures 8-11 were revised:*

Fig. 1 | The atomic structure of MS- $\text{Ti}_3\text{C}_2\text{T}_x$ and HF- $\text{Ti}_3\text{C}_2\text{T}_x$. The HAADF-STEM images with low magnification of the FIB milling slices of (a) pristine MS-MXene and (b) pristine HF-MXene. The bright white ribbons marked by white arrows are the regions with the appropriate zone axis which could be imaged in IDPC-STEM mode. c The HAADF-STEM image of layer-structured MS- $\text{Ti}_3\text{C}_2\text{T}_x$ with the d -spacing of 1.1 nm. Ti atoms are marked by blue dots. d The HAADF-STEM image of layer-structured HF- $\text{Ti}_3\text{C}_2\text{T}_x$ with the d -spacing of 1.3 nm. Ti atoms are marked by blue dots. e The IDPC-STEM image of MS- $\text{Ti}_3\text{C}_2\text{T}_x$ which is corresponding to (c). Ti atoms, C atoms and surface terminations are marked by blue, yellow and red dots, respectively. f The IDPC-STEM image of HF- $\text{Ti}_3\text{C}_2\text{T}_x$ which is corresponding to (d). Ti atoms, C atoms and surface terminations are marked by blue, yellow and red dots, respectively. g The edge spectra of STEM-EELS image of MS- $\text{Ti}_3\text{C}_2\text{T}_x$ indicating the existence of Ti, C, O and Cl signals. h HAADF-STEM image of the region terminated with $-\text{Cl}$ in MS- $\text{Ti}_3\text{C}_2\text{T}_x$.

Fig. 3 | Atomic-scale characterizations of ionic structures after electrochemical polarization. The HAADF-STEM and corresponding IDPC-STEM images of (a) MS-initial-lithiation and (b) cycled-MS-MXene. Ti atoms, C atoms, surface terminations and Li atoms are marked by blue, yellow, red and purple dots, respectively. The contrast of the terminations in MS-MXene is weak in HAADF images, but became clear in the corresponding IDPC images. c The EELS-STEM mapping of Ti, C, O and Li of MS-initial-lithiation sample. **d** A schematic showing the full desolvation in MS-Ti₃C₂T_x. **The HAADF-STEM and corresponding IDPC-STEM images of (e) HF-initial-lithiation and (f) cycled-HF-MXene. Ti atoms, C atoms, surface terminations and Li atoms are marked by blue, yellow, red and purple dots, respectively. g** EELS-STEM mapping of Ti, C, O and Li of cycled-HF-MXene sample. **h** A schematic showing the solvents co-intercalation in HF-Ti₃C₂T_x.

Supplementary Fig. 8. The HAADF-STEM images with low magnification of the FIB milling slices of MS-initial-lithiation, cycled-MS-MXene, HF-initial-lithiation and cycled-HF-MXene samples. The layers are bent in certain areas induced by FIB milling. The bright white ribbons marked by white arrows are the regions with the appropriate zone axis which could be imaged in IDPC-STEM mode.

Supplementary Fig. 9. Atomic-scale characterizations of ionic structures after electrochemical polarization of MS-MXenes. The HAADF-STEM images and corresponding IDPC-STEM images at the lower magnification with different regions of (a) MS-initial-lithiation and (b) cycled-MS-MXene suggesting the consistency of interlayer ionic structures. Ti atoms, C atoms, surface terminations and Li atoms are marked by blue, yellow, red and purple dots, respectively.

Supplementary Fig. 11. Atomic-scale characterizations of ionic structures after electrochemical polarization of HF-MXenes. The HAADF-STEM images and corresponding IDPC-STEM images at the lower magnification of (a) cycled-HF-MXene and (b) HF-initial-lithiation suggesting the consistency of interlayer ionic structure. Ti atoms, C atoms, surface terminations and Li atoms are marked by blue, yellow, red and purple dots, respectively. (c) EELS-STEM mapping of Ti, C, O and Li of HF-initial-lithiation sample. Ti atoms are marked by blue dots.

2. Page 6, we have revised the statement:

“Figure 1a and 1b show the cross-section HAADF-STEM images of both pristine MS-Ti₃C₂T_x and HF-Ti₃C₂T_x MXene slices cut by focused ion beam (FIB) technique, exhibiting the expected layered structures. The layers are bent in certain areas, induced by FIB milling. The HAADF-STEM images at higher magnification show the atomic-scale layered structure of both MS-Ti₃C₂T_x and HF-Ti₃C₂T_x MXenes in Fig.1c and 1d, which contains three layers of titanium...”

3. Page 12, we have revised the statement:

“All samples were prepared by electrochemical polarization at a constant potential of 0.2 V (vs. Li⁺/Li) during the initial stage and after three CV cycles. The TEM samples need to be ion-milled to ultra-thin slices to minimize the projection effect. The low magnification cross-section HAADF-STEM images

of four lithiated MXene FIB samples are showed in Supplementary Fig. 8, exhibiting the layered structure. Figures 3a and 3b...

The additional individual white dots located in the middle of interlayers, highlighted by a purple dashed frame, are clearly resolved in IDPC-STEM images but cannot be seen in the corresponding HAADF-STEM images in Fig. 3a and 3b. The HAADF-STEM and corresponding IDPC-STEM images in other larger regions of lithiated samples were showcased in Supplementary Fig. 9.”

4. Page 15, we have revised the statement:

“Importantly, the presence of solvent molecules between HF-Ti₃C₂T_x interlayers (noted as irregular white blocks in Fig. 3e and Fig. 3f) is now clearly visible from IDPC-STEM images, while the conventional HAADF-STEM falls short in detecting these light elements. The HAADF-STEM and corresponding IDPC-STEM images of continuous multi-layer regions (Supplementary Fig. 11a and 11b) demonstrated consistent *d*-spacing and ionic structures. The interlayer ionic structure is further established by mapping the distribution of Ti atoms, carbon (C) atoms, O atoms and Li ions.”

5. Page 22, we have added the details of samples preparation in Methods:

“The IDPC and HAADF images were conducted in spherical aberration-corrected TEM instruments (Spectra 300 X-CFEG (FEI) operated at 300kV, ARM200F (JEOL) operated at 200kV), and EELS spectra were acquired on the Gatan 965 GIF QUANTUM ER with the GIF Continuum K3 camera. The samples for STEM imaging were prepared by an Thermo Scientific Scios 2 DualBeam Focused Ion Beam (FIB) system.”

Specific Comment 2: *Although contrast can be observed in the interlayer and elements can be detected by EELS, how do the author obtain one-to-one correspondence for proposed atomic model and the image. For example, the weak interlayered contrast in the middle of Fig 3b was assigned to be Li atoms. Why could not be oxygen atoms, or at least partially oxygen atoms? Besides, the signals of O and Li in EELS are almost in the level of noise. So I could not agree with the authors strong statements of direct atomic-scale observation of full desolvated Li ions or the embedded “solvent” layer.*

Response:

We thank the Referee for the important comment. We would like to emphasize first that the evidence of desolvated and solvated Li intercalation in MS-MXene and HF-MXene, respectively, is not only based on STEM analysis but on a combination of different techniques including NMR and in-situ-XRD, that all support these observations.

Following Referee's comments, we conducted additional EELS-STEM characterizations using equipment with higher energy resolution and collected complementary data. Firstly, the comparison between IDPC-STEM images of pristine MXenes and lithiated MXenes show the presence of an extra layer of individual atoms in lithiated samples, assumed to be Li ions, as shown in the Figure R5 below. Secondly, the EELS-STEM mapping clearly show Li ions distributed in the middle of the interlayer, as can be seen in the new Figure 3 shown above. Since the samples were ultra-thin, the signal of Li is not expected to be strong. In addition, the IDPC images in lithiated MS-MXene demonstrate the presence of individual atoms (white dots marked by purple boxes) in the interlayers, and did not show any contrast due to solvent molecules (irregular white blocks marked by green lines), differently from what is shown in HF-MXene. The comparison between lithiated MS-MXene and HF-MXene is shown here as Figure R6, that clearly shows the absence of solvent in the interlayer of MS-MXene. We then collected new EELS-STEM edge spectra with higher resolution, whose signals are all obvious and not in the level of noise, as shown in Figure R7. The O and Li signals are all identifiable and the obvious extra Li peak could be detected in lithiated MS-MXene compared with pristine MS-MXene. Other EELS-STEM edge spectra with higher resolution was also provided in Figure 1 and Supplementary Figure 12 in the revised manuscript (see below). Last but not least, our other experimental results including in-situ-XRD and ^7Li ssNMR all agree with this conclusion. The *d*-spacing of MS-MXene stays constant during cycling via in-situ-XRD tests, indicating bare ion intercalation. And the shift of the intercalated Li ions peak from 22 ppm to 45 ppm via ^7Li ssNMR tests indicates the stronger confinement effect of Li ions in MS-MXene than that in HF-MXene.

Fig. R5: The comparison between the pristine MS-MXene and lithiated MS-MXene samples in IDPC-STEM mode. Ti atoms, C atoms, surface terminations and Li atoms are marked by blue, yellow, red and purple dots, respectively. These Figures were added in Figure 3 in the revised manuscript.

Fig. R6: The comparison between the lithiated MS-MXene and lithiated HF-MXene samples in IDPC-STEM mode. Ti atoms, C atoms, surface terminations and Li atoms are marked by blue, yellow, red and purple dots, respectively. These Figures were added in Figure 3 in the revised manuscript.

Fig. R7: The EELS-STEM edge spectra of pristine and lithiated MS-MXene with higher resolution. **These figures were added in Figure 1 and Supplementary Figure 12 in the revised manuscript.**

Changes to the manuscript:

1. Figures 1, 3 and Supplementary Figures 11 were revised (see above in the Changes to the manuscript on page 5).
2. Supplementary Figures 10 and 12 were revised:

Supplementary Fig. 10. Atomic-scale characterizations of ionic structures after electrochemical polarization. **(a)** EELS-STEM mapping of Ti, C, O and Li of cycled-MS-MXene sample. Ti atoms and surface terminations are marked by blue and red dots, respectively. HAADF-STEM and IDPC-STEM showcases the 2D projection of the region. As long as there is one Cl atom, the contrast of O terminations will be covered by Cl. **(b)** The intensity profiles along the direction indicated by the white arrow shown in IDPC-STEM image of cycled-MS-MXene sample. Ti atoms, C atoms, surface terminations and Li atoms are marked by blue, yellow, red and purple dots, respectively.

Supplementary Fig. 12. Edge spectra of STEM-EELS images of (a) lithiated MS-Ti₃C₂T_x and (b) HF-Ti₃C₂T_x indicating the existence of Li signals in the interlayers.

Specific Comment 3: *The authors mentioned that “The stability of the SEI layer on MS-MXene after cycling is confirmed by Fig. 4e, which shows almost no change with the initial SEI layer formed at the first cycle.” It is difficult to observe that there was no change in the morphology of the SEI before and after cycling. Before cycling, the SEI surface has shallow contrast particles, while after cycling, the SEI surface forms a shallow contrast film with a thickness of approximately 100 nm. In addition, the SEI image before cycling should be included in main text for better comparison.*

Response:

We appreciate the constructive comment and we apologize for the confusion. This sentence aimed at highlighting stability of SEI layer formed on MS-MXene, as the thickness of the SEI layer in MS-MXene didn't show significant degradation after cycling as illustrated in the new Figure R8 below. Moreover, TOF-SIMS, XPS, and FTIR characterizations reveal a decrease in the organic component content and an increase in the inorganic components (Li₂CO₃ and LiF) upon cycling, indicating that the SEI layer on MS-MXene becomes more robust with cycling. This statement has been revised in the manuscript.

Moreover, we reorganized the EDS mappings and showed the EDS mapping of MS-MXenes before and after cycling in Figure 4 and the EDS mapping of HF-MXenes before and after cycling in Supplementary Figure 14, as requested by the Referee.

Fig. R8: The EDS mapping of the SEI layers of lithiated MS-MXene samples. These Figures were added as Figure 4e and f in the revised manuscript.

Changes to the manuscript:

1. Page 18, we have updated the statement:

“The stability of the SEI layer on MS-MXene after cycling is confirmed by Fig. 4f, which shows almost no change in the thickness with the initial SEI layer formed at the first cycle.”

2. Figure 4 and Supplementary Figure 14 have been updated:

Fig. 4 | Characterizations of solvents and SEI layers. a-d TPD-MS measurement of DMC component ($m/z = 45$) of **(a)** HF-Ti₃C₂T_x electrodes polarized at 1.1V and further vacuum dried for 80°C, **(b)** MS-Ti₃C₂T_x electrodes polarized at 1.1V and further vacuum dried for 80°C, **(c)** expanded graph of vacuum dried HF electrodes and **(d)** expanded graph of vacuum dried MS electrodes and pristine MS powder. **EDS mapping of the cross section of (e) MS-initial-lithiation and (f) cycled-MS-MXene illustrating the cycling stability of the SEI layers.** **g-i** 3D reconstructions of different components in SEI layers via TOF-SIMS of **(g)** the initial cycle via CV test of MS-Ti₃C₂T_x, **(h)** the third cycle via CV test of MS-Ti₃C₂T_x, and **(i)** the initial cycle via CV test of HF-Ti₃C₂T_x.

Supplementary Fig. 14. EDS mapping of the cross section of (a) HF-initial-lithiation and (b) cycled-HF-MXene illustrating the degradation of SEI layers.

Specific Comment 4: Fig. 2b showed the b values were inconsistent with the descriptions in the text “the b value obtained from Equation (see methods) for MS- $Ti_3C_2T_x$ is larger than that of HF- $Ti_3C_2T_x$ (of 0.91 vs 0.71, respectively) in the range of 0.5 to 100 $mV s^{-1}$.”

Response: We have checked the data but we did not find any error nor inconsistency in the Figure 2b and the text. However, the confusion may arise from the black ($b = 1$) and orange ($b = 0.5$) line plots that represents the extreme values of b : $b = 0.5$ stands for a semi-infinite diffusion-controlled charge storage process, and $b = 1$ means a surface-controlled process. The b value of MS-MXene ($b = 0.91$) was close to 1, indicating the faster charge and ion transfer kinetics than that of HF-MXene ($b = 0.71$). For sake of clarity, we added two fitting lines showing the experimental b values (see Figure R9 below).

Fig. R9: The b values of MS- $Ti_3C_2T_x$ (0.91) and HF- $Ti_3C_2T_x$ (0.71) in the range of 0.5 to 100 $mV s^{-1}$. This Figure is now Figure 2b in the revised manuscript.

Changes to the manuscript:

1. We have revised the Figure 2b:

Fig. 2 | The electrochemical properties of MS-Ti₃C₂T_x and HF-Ti₃C₂T_x. **a** CV tests of MS-Ti₃C₂T_x and HF-Ti₃C₂T_x at the scan rate of 0.1 mV s⁻¹ in 1 M LiPF₆ (in 1:1 ethylene carbonate/dimethyl carbonate) electrolyte. **b** The b values of MS-Ti₃C₂T_x and HF-Ti₃C₂T_x in the range of 0.5 to 100 mV s⁻¹. **c** In-situ-XRD maps of (002) peaks of MS-Ti₃C₂T_x during GCD tests. **d** In-situ-XRD maps of (002) peaks of HF-Ti₃C₂T_x during GCD tests. ⁷Li ssNMR spectra of **(e)** HF-Ti₃C₂ samples lithiated in the initial and third cycle, and **(f)** MS-Ti₃C₂ samples lithiated in the initial and third cycle indicating the Li ions confined in between interlayers and free Li ions in SEI layers and absorbed at surface. The inset shows the comparison between the integrated areas of the peaks for intercalated Li ions in MS-Ti₃C₂ (purple region) and HF-Ti₃C₂ (orange region), illustrating more Li ions inserted in MS-Ti₃C₂ than HF-Ti₃C₂.

Reviewer #2 (Remarks to the Author)

This manuscript systematically investigates the ion desolvation and solvent-ions co-intercalation behaviors of HF-Ti₃C₂ and MS-Ti₃C₂ MXene with different surface chemistries during lithium-ion intercalation by utilizing advanced characterization tools such as IDPC-STEM, ssNMR, and TPD-MS. It is found that the O- and Cl-terminated MS-MXene can achieve complete desolvation intercalation of Li ions and form a stable SEI layer, resulting in increased capacity and power performance. This research is of great significance for understanding the role of ion desolvation in energy storage. However, several critical issues should be addressed before the manuscript can be considered for publication in Nature Communications:

General response: We appreciate the positive comments of the Referee regarding the novelty and importance of our work. We would like also to thank the Referee for their valuable comments, which help us to improve the quality of our manuscript. Following their comments, we have conducted additional EELS-STEM characterizations and XRD tests. We have also refined the Figures and added clarifying statements to improve the quality of the work. We provide below point-by-point answers to each of the points raised and outline the changes made in the revised manuscript. We trust that all the issues have been adequately resolved in line with your suggestions.

Specific Comment 1: *This manuscript proposes that O- and Cl-terminated MS-MXene allows complete desolvation intercalation of Li ions, whereas the F-, OH-, and O-terminated HF-MXene is accompanied by solvent co-intercalation. While the specific surface functional groups and their interactions with the lithium ions and solvents are undoubtedly important factors, other structural and compositional characteristics of the MXenes may also play a significant role. Are such differences simply due to different in surface chemistry, or do other factors also play a role?*

Response:

Thanks for the insightful comment. The intrinsic origin of charge storage in HF-MXene was demonstrated through DFT calculations in previous studies ([J. Mater. Chem. A, 2019,7, 16231-16238], [J Phys Chem Lett, 2018, 9, 1223-1228]). More specifically, the total density of states (DOS) near the Fermi level is primarily attributed to the d-orbitals of the titanium atoms. Therefore, the electron-

transfer process mainly happens on the Ti atom. When Ti atoms form chemical bonding with different terminals (i.e. MXenes with different terminations), the DOS of Ti will change, leading to the change of charge storage (see for instance: [J. Am. Chem. Soc., 2012, 134, 16909-16916], [J. Am. Chem. Soc., 2014, 136, 6385-6394]). In conclusion, surface terminations are assumed to play a dominant role in charge storage performance. However, those simulations do not take into consideration the electrolyte solvents and the nature of the SEI layer formed on the MXene. Thus, we focused on the intrinsic factor of surface chemistries and explore the effect of them on energy storages in this work. In response, we added the statements in the Introduction for the better illustration.

Changes to the manuscript:

1. Page 4, we updated the following statement in the Introduction:

“MXenes are usually prepared by etching the “A” element of MAX phase (A=early transition metal) in HF-containing electrolytes, resulting in the preparation of F-, OH- and O-terminated MXenes. Previous theoretical studies have shown the dominant role of the surface chemistry of HF-MXenes on lithium ions storage mechanism [24-27]. Notably, first-principles calculations predicted that the desolvation of cations would enhance their interaction with surface terminations in MXenes and charge transfer, resulting in improved energy storage performance [28]. In 2019, the intercalation of Li ions in confined interlayer space of F-terminated Ti_3C_2 MXene was reported, resulting in an increased capacity by fast redox charge transfer.”

2. Page 26, we have added the following references:

[24] Wang, L. et al. Origin of theoretical pseudocapacitance of two-dimensional supercapacitor electrodes $Ti_3C_2T_2$ (T = bare, O, S). *J. Mater. Chem. A* **7**, 16231-16238 (2019).

[25] Zhan, C. et al. Understanding the MXene pseudocapacitance. *J Phys Chem Lett* **9**, 1223-1228 (2018).

[26] Xie, Y. et al. Role of surface structure on Li-ion energy storage capacity of two-dimensional transition-metal carbides. *J. Am. Chem. Soc.* **136**, 6385-6394 (2014).

[27] Tang, Q., Zhou, Z. & Shen, P. Are MXenes promising anode materials for Li ion batteries? Computational studies on electronic properties and Li storage capability of Ti_3C_2 and $Ti_3C_2X_2$ (X = F, OH) monolayer. *J. Am. Chem. Soc.* **134**, 16909-16916 (2012).

[28] Ando, Y., Okubo, M., Yamada, A. & Otani, M. Capacitive versus pseudocapacitive storage in MXene. *Adv. Funct. Mater.* **30**, 2000820 (2020).”

Specific Comment 2: *When preparing HF- and MS-MXene samples for ^7Li ssNMR spectroscopy experiments, how to control the washing away of the surface electrolyte without affecting the solvents between interlayers in the MXene?*

Response:

This is an important point that we did not emphasize enough in the original version of the manuscript. We conducted XRD tests for the lithiated HF-MXene samples with and without DMC washing, such as shown in the Figure R10 below. The d -spacing stays almost constant, indicating DMC washing did not affect the presence of solvent molecules in the interlayers; we added this image as Supplementary Figure 7a in the revised manuscript. Moreover, the negligible P signals of LiPF_6 in SEI layers of all the lithiated MXene samples obtained via TOF-SIMS measurements indicated that DMC washing effectively removes the electrolyte on the surface of the electrodes as shown in Figure R11. We also added these images to Supplementary Figures 17 and 18 in the revised manuscript. In addition, we added the details of DMC washing in Method.

Fig. R10: XRD measurements on cycled-HF-MXene samples with and without DMC washing. **This Figure was added as Supplementary Figure 7a in the revised manuscript.**

Fig. R11: 3D reconstructions of P signals in SEI layers in three conditions via TOF-SIMS, after washing the MXenes in DMC solvent. **These Figures were added as Supplementary Figure 18 in the revised manuscript.**

Changes to the manuscript:

1. Page 22, we added the following sentence in the Method section:

“Four lithiated samples were immersed in the DMC with gentle shaking for a few seconds to remove the LP30 on the MXene surface, followed by drying overnight in vacuum.”

2. Page 11, we added the following sentence:

“The lithiated electrodes were then washed by DMC to remove the LP30 electrolytes on the surface and the same weight of materials was introduced in the rotors for NMR experiments. The XRD measurements shown in Supplementary Fig. 7a confirm that washing with DMC has a negligible effect on the solvents accommodated in the interlayers. These samples are further referred to as MS-initial-cycle, MS-third-cycle...”

3. We have added the XRD results in Supplementary Figure 7:

Supplementary Fig. 7. (a) XRD measurements on cycled-HF-MXene samples with and without DMC washing. These two samples were all dried in vacuum overnight. The d -spacing almost stayed constant indicating that DMC washing will not affect the solvents in the interlayers. ^7Li ssNMR spectra of (b) pristine MS-MXene and (c) pristine HF-MXene powders. No Li peaks before Li ions insertion in MS-MXene. As for HF-MXene, the slight peak that appears at 23 ppm (black frame) could be assigned to few Li ions inserted in between interlayers

during the LiF + HCl etching process. Notably, the intensity of this weak peak is much lower than the peak for intercalated Li ions for lithiated HF-MXene samples, illustrating the influence of pre-inserted Li ions is negligible.

4. We have added the detection of P signals in Supplementary Figure 17 and 18 and the statement in the label:

Supplementary Fig. 17. Depth profiles in negative mode of the SEI layers in three conditions via TOF-SIMS. The contents of organic components and inorganic components exhibit different evolutions when the depth increases. **The negligible signals of P indicated that washing by DMC could effectively remove the electrolytes in the surface of the lithiated MXene electrodes.**

Supplementary Fig. 18. 3D reconstructions of different components in SEI layers in three conditions via TOF-

SIMS. (a) Initial cycle via CV test of MS-Ti₃C₂T_x. **(b)** The third cycle via CV test of MS-Ti₃C₂T_x. **(c)** Initial cycle via CV test of HF-Ti₃C₂T_x. Organic components mainly distribute in the outer layer of SEI in MS-Ti₃C₂T_x and HF-Ti₃C₂T_x. The contents of F-contained inorganic components and Li₂CO₃ of SEI in MS-Ti₃C₂T_x increase while the content of organic components decreases after cycling. Less O-contained inorganic components are detected in HF-Ti₃C₂T_x than MS-Ti₃C₂T_x. **The negligible signals of P indicated that washing by DMC could effectively remove the electrolytes in the surface of the lithiated MXene electrodes.**

Specific Comment 3: *It is recommended to provide higher resolution EELS-STEM mapping and EDS mapping of SEI layers.*

Response:

We thank the Referee for the important comment. We conducted additional EELS-STEM characterizations on the pristine and lithiated MXene samples using an equipment with higher resolution (Gatan 965 GIF QUANTUM ER with the GIF Continuum K3 camera); EELS-STEM edge spectra and mapping images with higher resolution than previous results were then obtained. Taking the MS-initial-lithiation sample as example, the atomic structure of MS-MXene can be clearly detected (Figure R12), showing the boundaries of the interlayers. High resolution EELS-STEM mapping using the new equipment now shows that the distributions of Li and O signals are similar to the previous ones, clearly showing the presence of Li ions in the interlayer without solvent molecules (no C signal). In the revised manuscript, we added the EELS-STEM mapping of cycled-MS-MXene sample as Supplementary Figure 10a and all the EELS-STEM images (Figures 1g, 3c, 3g and Supplementary Figures 11c and 12) were replaced. We also added the details of EELS-STEM in the Methods section. Moreover, the SEI layer is quite sensitive to electron beam. We obtained the EDS mapping of SEI layer before it was damaged. Nevertheless, the counts for the EDS mapping are tens of thousands and they are the highest resolution image we could obtain, which is enough to identify the distributions of the various elements at this magnification.

Fig. R12: The EELS-STEM mapping of Ti, C, O and Li of MS-initial-lithiation sample. Ti atoms and terminations are marked as blue and red dots, respectively. **This Figure was added in Figure 3c in the revised manuscript.**

Changes to the manuscript:

1. *Figures 1g, 3c, 3g and Supplementary Figures 10a and 11c were revised (see above in the Changes to the manuscript on page 5 and 12).*

2. *Page 22 in the revised Manuscript, we have added the details of EELS-STEM characterization in Methods:*

“The IDPC and HAADF images were conducted in spherical aberration-corrected TEM instruments (Spectra 300 X-CFEG (FEI) operated at 300kV, ARM200F (JEOL) operated at 200kV), and EELS spectra were acquired on the Gatan 965 GIF QUANTUM ER with the GIF Continuum K3 camera.”

Specific Comment 4: *Whether different electrolytes interact with the -O or -F terminations on the surface of MXene and ultimately affect the formation of SEI and charge storage capacity in this work. The following literature can be consulted (Energy Environ. Sci., 2024,17, 3021-3031).*

Response:

We thank the Referee for quoting this recent work that we carefully analyzed (Energy Environ. Sci., 2024,17, 3021-3031). In that paper, it was shown that LiF and Li₂CO₃ of the SEI layer were generated from the oxidation decomposition of the carbonate electrolyte, and the content of inorganic components in the SEI layer was found to be increased after cycling. It was further proposed that the enrichment of the SEI layer in inorganic compounds with high ion conductivity could accelerate the

ion transfer and ion desolvation, which is in line with our work and previous studies ([J. Electrochem. Soc., 2024, 171, 030512], [Nat. Energy, 2020, 5 386-397], [Joule, 2019, 3, 2322-2333]). As suggested, we have cited the suggested important paper and added the relevant discussions into the main text.

Changes to the manuscript:

1. Page 19, we have updated the following statement:

“Combined with other characterization techniques, the analysis of the SEI layer shows that a ~200 nm-thick dense and stable SEI layer is formed on the surface of MS-MXene, enriched with inorganic components. The presence of such **thick and robust SEI layer, enriched with inorganic components with high conductivity, could facilitate the desolvation of the Li ions and their transfer through the electrochemical interface. The Li ion desolvation resulted in improved charge transfer, increased capacity and long-term stability of the electrodes by preventing the exfoliation of MXene interlayers** [48, 52, 53]. It is similar to what is observed in graphite negative electrodes of Li ion batteries.”

2. Page 30 in the revised Manuscript, we have added the following article into References:

“[48] Sunny, S., Coppel, Y., Taberna, P. L. & Simon, P. Characterization by NMR spectroscopy of the SEI layer formed on Ti₃C₂ MXene materials prepared with various terminations. *J. Electrochem. Soc.* **171**, 030512 (2024).

[52] Zhang, A. et al. Regulating electrode/electrolyte interfacial chemistry enables 4.6 V ultra-stable fast charging of commercial LiCoO₂. *Energy Environ. Sci.* **17**, 3021-3031 (2024).

[53] Heiskanen, S. K., Kim, J. & Lucht, B. L. Generation and evolution of the solid electrolyte interphase of lithium-ion batteries. *Joule* **3**, 2322-2333 (2019).”

Specific Comment 5: *Although the experimental data and characterization results support the conclusions, and the analysis and discussion are relatively comprehensive. However, there are some contents that can be further explored, such as the formation mechanism of SEI layer and the specific effect of the desolvation process on the electrochemical performance.*

Response:

The formation mechanism of the SEI layer is indeed important, and this is why previous work from some authors of the paper focused onto the SEI formation mechanism of MXenes with different

surface terminations by NMR characterizations (J. Electrochem. Soc., 2024, 171, 030512). In that paper, we showed that the presence of single bonded oxygen in MS-MXene led to organic lithium alkyl carbonate formation, which eventually converts to Li_2CO_3 . For the HF-MXene with F-terminations, LiF was generated in the SEI layers. The findings of this manuscript align well with those results and with previous studies ([Energy Environ. Sci., 2024,17, 3021-3031], [Nat. Energy, 2020, 5 386-397], [Joule, 2019, 3, 2322-2333]). As suggested, we have updated the following sections to further illustrate these points.

Changes to the manuscript:

1. *Page 18, we have updated the following statement:*

“The SEI formed on MS-MXene is mainly composed of O-containing inorganic (Li_2CO_3) components generated by the reaction between O-terminations and carbonate solvents, and the organic components in the outer layer degrade after cycling. Differently, the inorganic phase of the SEI layer on HF-MXene contains mainly F-based products resulting from the reaction between F-terminations and electrolytes with more organic components in the outer layer, in agreement with our previous results on the SEI layer formation [48].”

2. *Page 19, we have updated the following statement:*

“Combined with other characterization techniques, the analysis of the SEI layer shows that a ~200 nm-thick dense and stable SEI layer is formed on the surface of MS-MXene, enriched with inorganic components. The presence of such thick and robust SEI layer, enriched with inorganic components with high conductivity, could facilitate the desolvation of the Li ions and their transfer through the electrochemical interface. The Li ion desolvation resulted in improved charge transfer, increased capacity and long-term stability of the electrodes by preventing the exfoliation of MXene interlayers [48, 52, 53]. It is similar to what is observed in graphite negative electrodes of Li ion batteries.”

3. *Page 30, we have added the Refs. 48, 52 and 53 into References (see above in the Changes to the manuscript on page 25).*

Specific Comment 6: *Figures need to be further standardized.*

Response: We appreciate you for thoughtful comments. We have standardized the format of all the images reaching for the high quality of *Nature Communications*.

Specific Comment 7: *The format of references is not uniform, e.g., the journal name is the full title in ref.38.*

Response: We appreciate you for careful checks. We apologize for our mistakes. We have corrected the error and conducted the comprehensive review of the manuscript to prevent such mistakes.

Changes to the manuscript:

Page 29, the References list has been revised: “

[40] Xiang, Y. et al. Quantitatively analyzing the failure processes of rechargeable Li metal batteries. *Sci. Adv.* **7**, eabj3423 (2021).

[42] Li, Q. et al. Sieving carbons promise practical anodes with extensible low-potential plateaus for sodium batteries. *Natl. Sci. Rev.* **9** (2022).”

Reviewer #3 (Remarks to the Author)

The paper provides an in-depth investigation into the impact of ion desolvation on the charge storage performance of Ti_3C_2 MXene materials with varying surface chemistries, using a combination of experimental characterizations and molecular dynamics simulations. The authors successfully elucidate the relationship between ion-electrode interactions and surface terminations, demonstrating how desolvation can significantly enhance charge storage capacity. The study offers a comprehensive understanding of how tailoring surface functionalities can lead to the development of high-performance energy storage devices.

It highlights the advantages of O- and Cl-terminated MXenes in achieving complete ion desolvation and stable SEI formation, which contribute to enhanced electrochemical performance. The findings emphasize the critical role of surface chemistry in influencing ion transport and charge storage mechanisms. Overall, this work serves as a valuable resource for researchers in the field, providing new insights into the design principles of MXene-based energy storage materials. Before publication, the reviewer kindly recommends that there are several points needed to be addressed in revised manuscript.

General response: We sincerely thank the Referee for their positive comments that helped us to improve the quality of our manuscript. As requested, we have conducted additional EELS-STEM characterizations with higher resolution equipment. Additional details on the charge storage mechanism, SEI, and experimental procedures have been included. Below, we offer point-by-point responses to each comment. All revisions in the manuscript have been highlighted, and we hope that these changes address all the issues raised in your feedback.

Specific Comment 1: *In the Introduction section, it discusses the importance of understanding electrolyte ion dynamics and differentiates between capacitive (non-Faradaic) and battery-like (Faradaic) processes. However, the transition between these two processes is not clearly explained, and the reader may find it difficult to understand the relationship between ion desolvation and the resulting pseudocapacitive behavior. It is recommended to include a brief explanation or diagram to illustrate how ion desolvation influences the charge storage mechanism in MXenes. There should be a clarification of the charge storage mechanisms in the introduction.*

Response:

We understand the concern of the Referee as we only provide a summary of those concepts. The reason is that detailed explanations about the transition between capacitive and battery-like processes are provided in a paper published in 2022 as shown in Figure R13 below (Nat. Energy, 2022, 7, 222-228). This paper describes the transition between capacitive and battery-like processes as a result of the change in ion solvation during adsorption or intercalation in nanoconfined spaces, nanosized pores or interlayer. The more the ion desolvation, the closer it is to the surface of the host material and the stronger the ion/host interaction, resulting in partial charge transfer and increased charge density. This is the region where the so-called “pseudocapacitive” processes are observed.

Fig. R13: Pseudocapacitive processes can lie in the transition region. Region II is the transition region. (Nat. Energy, 2022, 7, 222-228).

In addition, previous studies demonstrated a significant enhancement in electrochemical charge storage performance due to ion desolvation predicted by first-principles calculations (Adv. Funct. Mater. 2020, 30, 2000820). More specifically, the desolvation of ions would enhance the interaction with O-containing terminations resulting in increased charge transfer and improved capacity. In this manuscript, we reported the highest capacity for O- and Cl- terminated MS-MXene (873 C g^{-1} at 0.1 mV s^{-1}) with the intercalation of fully desolvated Li ions, while F-, O- and OH- terminated HF-MXene resulted in the intercalation of solvated Li ions with smaller capacity (286 C g^{-1} at 0.1 mV s^{-1}). We experimentally verified that the surface terminations and the SEI layers were driving the ion desolvation and could improve the charge storage capacity, in line with our previous results ([J. Electrochem. Soc., 2024, 171, 030512], [Nat. Energy, 2022, 7, 222-228]).

As recommended by the Referee, we have updated some statements in the Introduction.

Changes to the manuscript:

1. Page 4, we updated the following statement in the Introduction:

“MXenes are usually prepared by etching the “A” element of MAX phase (A=early transition metal) in HF-containing electrolytes, resulting in the preparation of F-, OH- and O-terminated MXenes. Previous theoretical studies have shown that the surface chemistry of HF-MXenes play a dominant role in lithium ions storage [24-27]. Notably, first-principles calculations predicted that the desolvation of cations would enhance their interaction with surface terminations in MXenes, resulting in improved charge transfer and capacity [28]. In 2019, the intercalation of Li ions in confined interlayer space of F-terminated Ti_3C_2 MXene was reported, resulting in an increased capacity by fast redox charge transfer.”

2. Page 27, we have added Refs. 24-28 into References (see above in the Changes to the manuscript on page 19).

Specific Comment 2: *In Figure 1 (a-c), it is represented the same materials (MS- $Ti_3C_2T_x$ with the d-spacing of 1.1 nm), but it seems like Ti atoms which is presented as blue dots are marked differently. If so, there should be a specific explanation why there is a difference between the atomic structures for same materials.*

Response: We apologize for the different marks we used. We have now checked all the Figures and standardized the marks of different atoms (Ti, C, O/F and Li). The Ti atoms are the same for MS-MXene in Figures 1.

Specific Comment 3: *In Figure 1 (e), the intensity of the C and O elements in the STEM-EELS image is too weak to clearly distinguish these elements. To enhance the clarity and provide more evidence, it is recommended to include additional supplementary information regarding the presence and distribution of these elements.*

Response:

We thank the Referee for the important comments shared by the other Referees. Following the comments, we conducted additional EELS-STEM characterizations using an equipment with higher resolution (Gatan 965 GIF QUANTUM ER with the GIF Continuum K3 camera). EELS-STEM edge

spectra with high resolution were obtained to further verify the contents of different terminations of pristine MS-MXene powders. The identification of the different elements is based on the standard EELS Atlas in the DigitalMicrograph for EELS measurements. Figure R14 below shows the obvious presence of Ti (around 465 eV), C (around 28 eV), O (around 21 eV) signals with slight peak of Cl (around 210 eV), suggesting the presence of mainly O-terminations, consistent with the results of XPS, EDS and TPD-MS of pristine MS-MXene powders. Figure 1g was replaced in the revised manuscript with Figure R14 to address Referee's comment. We also added the statements in the main context and the details of EELS-STEM in Methods.

Fig. R14: The edge spectra of STEM-EELS image of MS-Ti₃C₂T_x indicating the existence of Ti, C, O and Cl signals. **This Figure was added as Figure 1g in the revised manuscript.**

Changes to the manuscript:

1. *Figure 1 was revised:*

Fig. 1 | The atomic structure of MS- $\text{Ti}_3\text{C}_2\text{T}_x$ and HF- $\text{Ti}_3\text{C}_2\text{T}_x$. The HAADF-STEM images with low magnification of the FIB milling slices of (a) pristine MS-MXene and (b) pristine HF-MXene. The bright white ribbons marked by white arrows are the regions with the appropriate zone axis which could be imaged in IDPC-STEM mode. c The HAADF-STEM image of layer-structured MS- $\text{Ti}_3\text{C}_2\text{T}_x$ with the d -spacing of 1.1 nm. Ti atoms are marked by blue dots. d The HAADF-STEM image of layer-structured HF- $\text{Ti}_3\text{C}_2\text{T}_x$ with the d -spacing of 1.3 nm. Ti atoms are marked by blue dots. e The IDPC-STEM image of MS- $\text{Ti}_3\text{C}_2\text{T}_x$ which is corresponding to (c). Ti atoms, C atoms and surface terminations are marked by blue, yellow and red dots, respectively. f The IDPC-STEM image of HF- $\text{Ti}_3\text{C}_2\text{T}_x$ which is corresponding to (d). Ti atoms, C atoms and surface terminations are marked by blue, yellow and red dots, respectively. g The edge spectra of STEM-EELS image of MS- $\text{Ti}_3\text{C}_2\text{T}_x$ indicating the existence of Ti, C, O and Cl signals. h HAADF-STEM image of the region terminated with $-\text{Cl}$ in MS- $\text{Ti}_3\text{C}_2\text{T}_x$.

2. Page 7, we have added the statement:

“Electron energy loss spectroscopy (EELS) analysis was conducted on pristine MS- $\text{Ti}_3\text{C}_2\text{T}_x$, in which O signals (at 21 eV) and weak Cl signals (at 210 eV) were found in the edge spectra shown in Fig. 1g.”

3. Page 22, we have added the details of EELS-STEM characterization in Methods:

“The IDPC and HAADF images were conducted in spherical aberration-corrected TEM instruments (Spectra 300 X-CFEG (FEI) operated at 300kV, ARM200F (JEOL) operated at 200kV), and EELS

spectra were acquired on the Gatan 965 GIF QUANTUM ER with the GIF Continuum K3 camera.”

Specific Comment 4: *The author compares HF-Ti₃C₂ and MS-Ti₃C₂, emphasizing the role of different surface terminations. However, it would be beneficial to provide more justification for why these specific MXene surface chemistries were selected. Expanding on the rationale for using F-, OH-, O- (HF-Ti₃C₂) and O-, Cl- (MS-Ti₃C₂) terminations, and their expected impact on ion desolvation and intercalation, would strengthen the paper. There should be a justification for the selection of MXene surface chemistries.*

Response:

We appreciate the constructive feedback and apologize for not providing a more thorough explanation regarding the selection of these two MXenes. The origin of this work lies in a paper some of us published (Nat. Mater., 2020, 19, 894-899), where we reported a new synthesis route for MXene based on molten salt etching, alternative to the HF-based etching process only available at that time. As a result, we were able to prepare for the first time F-free, but O- and Cl- terminated MS-MXene. Surprisingly, the electrochemical behavior of O- and Cl- terminated MS-MXene during Li ion intercalation in non-aqueous electrolyte exhibited symmetric CV shape together with much higher capacity and high power performance compared to the conventional F-, O- and OH- terminated HF-MXene. That left, at that time, lot of questions regarding the origin of the different electrochemical behaviors, knowing that these materials only differed by their surface terminations of MXenes. However, we could only speculate that the change in electrochemical behavior as the results of different surface chemistry lies in the different degrees of ion desolvation through in-situ-XRD tests, because of a lack of direct evidence of this desolvation.

In this present work, we further explored the specific effect of surface chemistry on energy storage mechanisms by combining different analytical, structural and electrochemical techniques together with atomic-scaled characterization technologies, that allowed to get the full picture of the performance enhancement. As suggested, we have added some statements.

Changes to the manuscript:

1. Page 4, we updated the following statement in the Introduction:

“MXenes are usually prepared by etching the “A” element of MAX phase (A=early transition metal)

in HF-containing electrolytes, resulting in the preparation of F-, OH- and O-terminated MXenes. In previous theoretical studies, the surface chemistry of HF-MXenes play a dominant role in lithium ions storage [24-27]. Notably, the first-principles calculations predicted that the desolvation of cations would enhance their interaction with surface terminations in MXenes and charge transfer, resulting in the improved energy storage performance [28]. In 2019, the intercalation of Li ions in confined interlayer space of F-terminated Ti_3C_2 MXene was reported, resulting in an increased capacity by fast redox charge transfer.”

2. Page 4 in the revised Manuscript, we added the following statement:

“O- and Cl-terminated MS- Ti_3C_2 MXene exhibited high capacity (beyond 200 mAh g⁻¹) and high-power performance when used as the negative electrode during Li ion intercalation in carbonate-based electrolyte, while conventional HF etched Ti_3C_2 MXene terminated with F, O and OH exhibited a poor electrochemical behavior in the same condition. The results of in-situ-XRD suggested that different evolutions of *d*-spacing of two MXenes might be contributed to the different degrees of Li ion desolvation, which plays a significant role in pseudocapacitive processes. However, there is still a lack of direct evidence of this desolvation resulting in the ambiguity of how the terminations affect the ion desolvation. The reasons for this lie in the difficulty of obtaining an atomic resolution view of surface terminations...”

3. Page 27, we have added Refs. 24-28 into References (see above in the Changes to the manuscript on page 19).

Specific Comment 5: In the Methods section, Although the experimental methods are described, some details are missing, such as the specific conditions for the ion intercalation experiments and the justification for choosing particular characterization techniques like IDPC-STEM. Providing additional information on these points would improve the reproducibility and understanding of the experimental procedures.

Response:

The lithiated samples were polarized from OCV (about 2.7V vs Li^+/Li) down to 0.2 V (versus Li^+/Li) at the scan rate of 0.5 mV s⁻¹. Samples were kept at 0.2 V vs Li^+/Li until as steady state current was reached.

Regarding the sample preparation, we used DMC to wash the lithiated MXene electrodes during

few seconds to remove the electrolyte present on the MXene surface. The samples were then cut into slices using a Thermo Scientific Scios 2 DualBeam Focused Ion Beam (FIB) system before conducting HAADF-STEM, IDPC-STEM, EELS-STEM and EDS characterizations. In the revised manuscript, we have added more details in the Method for the better illustration.

Changes to the manuscript:

1. *Page 21, we have updated the following statement:*

“The lithiated samples for IDPC-STEM, HAADF-STEM, and SEI characterizations were polarized from OCV (about 2.7V versus Li^+/Li) down to 0.2 V (versus Li^+/Li) at the scan rate of 0.5 mV s^{-1} . Samples were kept at 0.2 V (versus Li^+/Li) until as steady state current was reach. Then, the coin cells were disassembled in a glove box and the lithiated MXene electrodes were immersed in the DMC with gentle shaking for a few seconds to remove the LP30 in the surface followed by drying overnight in vacuum.”

2. *Page 22, we have added the details of samples preparation in Methods:*

“The IDPC and HAADF images were conducted in spherical aberration-corrected TEM instruments (Spectra 300 X-CFEG (FEI) operated at 300kV, ARM200F (JEOL) operated at 200kV), and EELS spectra were acquired on the Gatan 965 GIF QUANTUM ER with the GIF Continuum K3 camera. The samples for STEM imaging were prepared by an Thermo Scientific Scios 2 DualBeam Focused Ion Beam (FIB) system.”

Specific Comment 6: *The paper reports the formation of a solid electrolyte interface (SEI) layer in MS- Ti_3C_2 , contributing to enhanced charge storage. It would be valuable to include a more detailed discussion on how the SEI layer's composition and thickness were determined and how these factors influence the long-term stability and performance of the MXene electrodes.*

Response:

EDS, TOF-SIMS and FTIR analysis showed that the thickness of the SEI layer formed on MS-MXene was kept constant after cycling. This thick, stable SEI layer was found to be rich in inorganic components with high ion conductivity thanks to the O-rich surface chemistry of MS-MXene, favoring ion desolvation and Li ions transfer through the electrochemical interface. The desolvation resulted in

improved charge transfer, increased capacity and long-term stability of the electrodes by preventing the exfoliation of MXene interlayers. All these results well match with previous work from the literature ([J. Electrochem. Soc., 2024, 171, 030512], [Nat. Energy, 2020, 5 386-397], [Joule, 2019, 3, 2322-2333], [Nat. Energy, 2022, 7, 222-228]). As suggested, we have updated the following statements for detailed discussions.

Changes to the manuscript:

1. *Page 18, we have updated the statement:*

“The stability of the SEI layer on MS-MXene after cycling is confirmed by Fig. 4e, which shows almost no change **in the thickness** versus the initial SEI layer formed at the first cycle.”

2. *Page 18, we have updated the following statement:*

“The SEI formed on MS-MXene is mainly composed of O-containing inorganic (Li_2CO_3) components **generated by the reaction between O-terminations and carbonate solvents**, and the organic components in the outer layer degrade after cycling. Differently, the inorganic phase of the SEI layer on HF-MXene contains mainly F-based products **resulting from the reaction between F-terminations and the electrolyte solvents** with more organic components in the outer layer, **in agreement with our previous results on the SEI layer formation** [48].”

3. *Page 19, we have updated the following statement:*

“Combined with other characterization techniques, the analysis of the SEI layer shows that a ~200 nm-thick dense and stable SEI layer is formed on the surface of MS-MXene, enriched with inorganic components. The presence of such **thick and robust SEI layer, enriched with inorganic components with high conductivity**, could facilitate the desolvation of the Li ions and their transfer through the electrochemical interface. **The Li ion desolvation resulted in improved charge transfer, increased capacity and long-term stability of the electrodes by preventing the exfoliation of MXene interlayers.** [48, 52, 53]. It is similar to what is observed in graphite negative electrodes of Li ion batteries.”

4. *Page 30, we have Refs. 48, 52 and 53 into References (see above in the Changes to the manuscript on page 25).*

Specific Comment 7: *The author mentions using molecular dynamics (MD) simulations to support the experimental findings. It would be helpful to provide more information on how these simulations were set up, including the specific models and parameters used. Additionally, discussing any limitations of the MD simulations and how they complement the experimental results would strengthen the discussion.*

Response:

We apologize for the lack of details and appreciate the opportunity to clarify the setup and parameters used in our molecular dynamics (MD) simulations and to discuss their limitations and complementary role to the experimental findings. We updated the details and discussions of MD simulations in revised Manuscript and Supplementary Information.

Changes to the manuscript:

1. Page 23, we have added more statements in Method:

“Supplementary Fig. 21 shows the molecular simulation system of MS-MXene and HF-MXene immersed into 1M LiPF₆ in EC/DMC electrolyte. The stoichiometric ratios of the terminations on the MS-MXene and HF-MXene surfaces were estimated based on experimental data. The distance between MS-MXene and HF-MXene layers were set according to the experimental XRD results. The Large-scale Atomic/Molecular Massively Parallel Simulator (LAMMPS) classical molecular dynamics code package are utilized for all simulations¹. The schematic atomic models were visualized and rendered using the OVITO software². The bonding and non-bonding interactions among the atoms are characterized utilizing the ClayFF force field³. The force field parameters, encompassing Lennard-Jones potential, point partial charges, harmonic bonds, and harmonic angles, are adopted from previous studies⁴⁻⁶. The particle-particle particle-mesh (PPPM) method was used in the k-space to handle long-range electrostatic interactions. The simulations were conducted in the Canonical ensemble (NVT) at a target temperature of 350 K, using an integration time step of 1 fs. To investigate the dynamic process of Li ions and solvents embedded between the MS-MXene and HF-MXene layers in the electrolyte upon charging, -2V potentials were applied to the MS-MXene and HF-MXene electrodes using the Constant Potential Method (CPM), respectively [59]. This CPM method maintains a constant potential difference between the electrodes and captures charge fluctuations on the electrode atoms and the ions dynamic of electrolytes throughout the simulation. The calculated radial distribution function of MXene surface terminations (F and O) with CH₃ from intercalated solvents in HF-MXene are shown

in Supplementary Fig. 22a and b. The calculated radial distribution function of intercalated Li ions with O terminations of MS-MXene and HF-MXene are displayed in Supplementary Fig. 22c.”

2. Page 17 in the revised Supplementary Information, we have added more details:

Supplementary Table 2. Lennard-Jones coefficients for the MS-MXene and HF-MXene layer with -F, =O, -OH, -Cl terminations.

Element	$\epsilon / \text{kcal.mol}^{-1}$	σ / nm
Ti	0.60870	0.19565
C	0.06600	0.35000
F	0.16730	0.31430
O(=O)	0.15540	0.31656
O(-OH)	0.18480	3.55320
H(-OH)	0.01000	0.90000
Cl	0.12470	3.78500

Specific Comment 8: *For the consistency throughout the paper, the term “Li+ ions” should be replaced to term “Li ions”.*

Response: We appreciate the careful checks. All the terms “Li+” have been replaced with “Li ions”. We have checked all the paper to make the consistency.

Specific Comment 9: *In Introduction section, it is needed to develop a consistency in terminology. The terms “pseudocapacitive” and “capacitive” are used to describe the charge storage mechanisms. For consistency and clarity, it is suggested to define these terms clearly in the Introduction and ensure their consistent use throughout the text.*

Response: We appreciate the important comments. In introduction, we first mention two extreme

energy storage processes: purely capacitive storage and purely Faradaic process. Then, the term “pseudocapacitive” is introduced to depict the continuous transition region between these two extreme cases, in agreement with Ref 7 (Nat. Energy, 2022, 7, 222-228). In addition, charge storage mechanism in MXenes being pseudocapacitive (see Ref 30, Li et al., Nat. Mater. 19, 894-899 2020), we now only use the term “pseudocapacitive” in the manuscript to be consistent.

Response to Referee Reports

Reviewer #1 (Remarks to the Author)

In the revised Figure 1g, the authors provide elemental detection using EELS. I can see an obvious mistake in this data. For oxygen element, the major edge is the K edge (~530 eV), while the minor edge is the L edge (~20 eV). It is contradictory that in the figure where signal from minor edge can be observed clearly whereas the major K edge signal can not be seen. It is doubtful whether the 20 eV peak really originates from oxygen. The authors should check their data carefully. Even if I assume the 20 eV signal is from oxygen. It is obvious the concentration of oxygen is much higher than lithium from many of the EELS spectra provided by the authors.

The authors did not respond to my comment² that how can it be excluded that the interlayer contrast is not oxygen, but solely lithium. The strong conclusion is not reliable. Moreover, I can also provide other explanations for such contrast. For example, as can be seen from low magnification images that the layer is heavily wavy. The authors also claim that they chose specific positions where the samples are in zone axis and clear images can be taken. Then I would propose that the interlayer contrast could be originated from the wavy nature of the layer, so that inclined portions can be existed in the pathway of the electron beam and therefore the 2D projection shows artifacts. In all, the improvements made by the authors still could not support their strong conclusion.

Response: We thank the referee for pointing this out. We fully agree with the Referee: the major K edge is around 530 eV and we have shown in our original manuscript the major K edge signal of oxygen, see the Figure below as a reminder, but the resolution of the EELS data was insufficient.

Original Figure 1e: EELS spectra in the original manuscript showing the main K edge signal of O at 530 eV (right).

Following the comments of the Referees made at the first round of revision, we carried out several additional experiments (including high resolution STEM and EELS) and modified the figures, and

more specifically those showing HAADF-, IDPC-STEM and EELS analysis (Figures 1 and 3). Then, during the rearrangement of Figure 1 to include new data, we mistakenly inserted the wrong image corresponding to the EELS O L edge at 20 eV. We have now redrawn Figure 1g based on the new data, highlighting all the characteristic peaks, as shown in Figure R1 below. The Figure R1 clearly evidences the presence of O.

This was our mistake, and we sincerely apologize for it. While the extensive modifications and the addition of new results in the revised version may have contributed to the issue, we should have more carefully checked the figures.

Figure R1: The edge spectra of STEM-EELS image of pristine MS-MXene indicating the existence of Ti, C, O and Cl signals. **This Figure was added as Figure 1g in the revised manuscript.**

Regarding Comment 2, we agree that the extra dots in the interlayer spacing may not all correspond to lithium atoms individually. However, we can first confirm the insertion of Li ions based on the EELS edge spectra (see Figure R2 below) that, combined with IDPC-STEM and ssNMR data, evidences that Li ions are inserted between the MS-MXene layers without the presence of solvents. Then, while it is still possible that the atomic columns are not perfectly aligned with the projection direction, in that case the additional dots should appear much brighter if they correspond to inclined titanium or oxygen atoms in the case of wavy MXene, as shown in the black frames in Figure R3. The intensity of these extra dots in Figure 3 in the manuscript is quite low, indicating that, although other atoms may be present, the majority of the contrast likely originates from Li ions. In fact, whether or not all the interlayer contrasts in the IDPC images are exclusively and 100% associated with Li ions does not alter our primary conclusion, which is the ion desolvation process. To diminish the misunderstanding, we have added the statements and updated the annotations of Figure 3 and Supplementary Figure 11 in the revised manuscript.

Figure R2: The edge spectrum of EELS-STEM of lithiated MS-MXene indicating the obvious Li peak. **This Figure was added in Supplementary Figure 12.**

Figure R3: The STEM image of the wavy MXene interlayers.

Changes to the manuscript:

1. *Figures 1 and 3 and Supplementary Figure 9, 11 and 12 were revised:*

Fig. 1 | The atomic structure of MS-Ti₃C₂T_x and HF-Ti₃C₂T_x. The HAADF-STEM images with low magnification of the FIB milling slices of (a) pristine MS-MXene and (b) pristine HF-MXene. The bright white ribbons marked by white arrows are the regions with the appropriate zone axis which could be imaged in IDPC-STEM mode. c The HAADF-STEM image of layer-structured MS-Ti₃C₂T_x with the *d*-spacing of 1.1 nm. Ti atoms are marked by blue dots. d The HAADF-STEM image of layer-structured HF-Ti₃C₂T_x with the *d*-spacing of 1.3 nm. Ti atoms are marked by blue dots. e The IDPC-STEM image of MS-Ti₃C₂T_x which is corresponding to (c). Ti atoms, C atoms and surface terminations are marked by blue, yellow and red dots, respectively. f The IDPC-STEM image of HF-Ti₃C₂T_x which is corresponding to (d). Ti atoms, C atoms and surface terminations are marked by blue, yellow and red dots, respectively. g The edge spectra of STEM-EELS image of MS-Ti₃C₂T_x indicating the existence of Ti, C, O and Cl signals. h HAADF-STEM image of the region terminated with -Cl in MS-Ti₃C₂T_x.

Fig. 3 | Atomic-scale characterizations of ionic structures after electrochemical polarization. The HAADF-STEM and corresponding IDPC-STEM images of (a) MS-initial-lithiation and (b) cycled-MS-MXene. Ti atoms, C atoms and surface terminations are marked by blue, yellow and red dots, respectively. The contrast of the terminations in MS-MXene is weak in HAADF images, but became clear in the corresponding IDPC images. c The EELS-STEM mapping of Ti, C, O and Li of MS-initial-lithiation sample. d A schematic showing the full desolvation in MS-Ti₃C₂T_x. The HAADF-STEM and corresponding IDPC-STEM images of (e) HF-initial-lithiation and (f) cycled-HF-MXene. Ti atoms, C atoms and surface terminations are marked by blue, yellow and red dots, respectively. g EELS-STEM mapping of Ti, C, O and Li of cycled-HF-MXene sample. h A schematic showing the solvents co-intercalation in HF-Ti₃C₂T_x.

Supplementary Fig. 9. Atomic-scale characterizations of ionic structures after electrochemical polarization of MS-MXenes. The HAADF-STEM images and corresponding IDPC-STEM images at the lower magnification with different regions of (a) MS-initial-lithiation and (b) cyclized-MS-MXene suggesting the consistency of interlayer ionic structures. Ti atoms, C atoms and surface terminations are marked by blue, yellow and red dots, respectively.

Supplementary Fig. 11. Atomic-scale characterizations of ionic structures after electrochemical polarization of HF-MXenes. The HAADF-STEM images and corresponding IDPC-STEM images at the lower magnification of (a) cycled-HF-MXene and (b) HF-initial-lithiation suggesting the consistency of interlayer ionic structure. Ti atoms, C atoms and surface terminations are marked by blue, yellow and red dots, respectively. (c) EELS-STEM mapping of Ti, C, O and Li of HF-initial-lithiation sample. Ti atoms are marked by blue dots.

Supplementary Fig. 12. Edge spectra of STEM-EELS images of (a) lithiated MS-Ti₃C₂T_x and (b) lithiated

HF-Ti₃C₂T_x indicating the existence of Li signals in the interlayers.

2. *Page 7, we have revised the statement:*

“Electron energy loss spectroscopy (EELS) analysis was conducted on pristine MS-Ti₃C₂T_x, in which O signals (at around 530 eV) and weak Cl signals (at around 207 eV) were found in the edge spectra shown in Fig. 1g.”

3. *Page 15, we have added the statement:*

“The corresponding edge spectra of the EELS-STEM images also support the existence of Li signals as shown in Supplementary Fig. 12. It is worth noting that although the extra dots in the interlayer spacing in Fig. 3a and b may not all correspond to lithium atoms individually, the majority of the contrast likely originates from Li ions since it is quite weak and only appear in IDPC-STEM mode.”

Reviewer #2 (Remarks to the Author)

The authors well considered my comments, now it is acceptable for publication now.

Response: We sincerely thank the Referee for the positive evaluation of our work, and we are also grateful for the Referee’s valuable comments, which help us to improve the quality of our manuscript.

Reviewer #3 (Remarks to the Author)

The authors have provided detailed responses and made necessary modifications to strengthen the overall quality and clarity of the paper. The authors’ efforts in improving the manuscript are commendable, and the current version is well-prepared for dissemination. As a result, the authors have thoroughly addressed all the concerns raised by the reviewer in the revised manuscript, and it is now recommended for publication.

Response: We sincerely thank the Referee for the recognition of our work, and we are also grateful for the Referee’s professional insights, which have improved the quality of our manuscript.

Response to Referee Reports

Reviewer #1 (Remarks to the Author)

In the last round of review, I did not see O-K signal in the middle of Fig1.g, as compared with Ti signal. However, in this updated document, the data has been changed. The O-K signal is comparable to Ti signal. Besides, the authors claimed that they have mistakenly used a wrong image in the manuscript in the last round of review. From these aspects, I'm quite confused with the data which is inconsistent and it seems the authors are not rigorous, which does not meet the criterion of Nature journals. For the identification of Li ions, in my personal view, the conclusion drawn from IDPC-STEM image contrast and weak EELS signal is far from enough. Whereas the interlayered Li is a critical point for this manuscript, therefore I could not be positive.

Response: We appreciate the Referee#1 for the careful check and comments. As previously mentioned, we made a mistake by using the wrong image in the first revision and we apologize again for that. However, we remind to the Referee that we extensively revised our manuscript in the first round, adding a lot of new results, resulting in the modification of about 15 Figures compared to the original version. The fact that this very specific figure (EELS analysis) was not updated and left in the revised paper is an error, for sure, but does not mean that “the authors are not rigorous, which does not meet the criterion of Nature Journals”.

Regarding the comment on the EELS signal, we did not expect the Li ion signal to be as strong as heavier elements, obviously, and new technology has to be developed to better display Li EELS signal. We believe that the Li signal we obtained is the best that can be achieved using the current technology. We would like to remind that the presence of Li ions in the interlayer of MXene is not only based on TEM and EELS analysis/images, but is also supported by NMR and electrochemistry. Indeed, only the intercalation of Li ions - as the sole cation present in the electrolyte - can explain the high capacity (200 mAh/g) achieved by MS-MXenes.

Finally, we added in the response to the Referee#4 new STEM images and many technical details and discussion (see for instance Figure R5), that all strengthen our conclusions; we hope that Referee#1 will find answers to their questions.

Reviewer #4 (Remarks to the Author)

First, I read the entire paper and found the results to be of high quality and significant importance. The characterizations presented (iDPC, EELS maps) are essential for supporting the main conclusions of the authors. I also recognize the high quality of the electron microscopy (EM) experiments on these extremely challenging samples. Therefore, the results should be evaluated carefully. However, I believe there are too many experimental unknowns for the paper to be ready for publication in its current form. Despite focusing on the EELS evaluation as requested, I also reviewed the iDPC images, as I am somewhat familiar with the technique, and the images are closely linked to the EELS maps/spectra.

Final Comments:

These questions are highly technical, but I believe the electron microscopy experiments are crucial for this publication. The standards of Nature Communications are exceptionally high, and the authors must provide sufficient information for other researchers to reproduce the experiments. Currently, I do not think there is enough detail to ensure reproducibility.

General response: We would like to thank Referee#4 for their positive comments and technical questions that helped us to improve the quality of the manuscript. Following these comments, experimental details and new STEM images were added in this revised version, and we provided as well point-by-point answers to each of the concerns raised. All the changes made to the revised manuscript have been highlighted in the main text. These important suggestions help us better highlight the reliability and reproducibility of the results of our STEM characterizations.

Specific Comment 1: Beam Sensitivity of MXenes

i. MXenes (including $Ti_3C_2T_x$) are extremely beam-sensitive, but this is not addressed in the paper.

This sensitivity could impact both the iDPC and EELS maps, especially at 300 kV.

ii. Experimental details, such as the beam current, are missing. For instance, was DCFI used?

Typically, dDPC is preferred—did the authors examine these images?

iii. What is the EELS energy dispersion? Was Fourier-log deconvolution applied?

Response:

We thank the Referee for the important comments. As MXenes are beam-sensitive, we tried many times to determine the appropriate experimental conditions at all steps, from FIB cutting to STEM

characterizations. The details are as followings.

The samples for STEM imaging were prepared by an Thermo Scientific Scios 2 DualBeam Focused Ion Beam (FIB) system. Low beam current of 48 pA (5 kV) and 43 pA (2 kV) were applied to thin the samples finely, minimizing the beam damage.

Then, the iDPC and HAADF images were collected using spherical-aberration-corrected TEM instruments (Spectra 300 X-CFEG (FEI) operated at 300 kV, and also ARM200F (JEOL) operated at 200 kV for comparison). We tested various beam currents, ranging from 0.1 nA to 5 pA. Typical images captured from the same sample in the adjacent area using different beam currents are shown in Figure R1. A beam current of 0.1 nA caused noticeable damage indicated by the white frame, while 5 pA was too weak to produce a clear image. Therefore, beam currents of 0.05 nA and 0.01 nA were used, and beam blank was set when not taking the images. We didn't perform DCFI nor dDPC. Within the range of electron dose that the MXene samples could withstand, higher doses were applied to acquire clear images.

Fig. R1: The HAADF and iDPC images of lithiated HF-MXene samples with beam currents of (a) 0.1 nA, (b) 0.05 nA (our experimental condition) and (c) 5 pA.

EELS spectra were acquired on the Gatan 965 GIF QUANTUM ER with the GIF Continuum K3 camera at 200 kV. The energy dispersion of EELS-STEM is 0.45 eV/Ch. Energy dispersion represents the energy resolution per pixel, i.e., the width of each energy loss signal, with units in eV/Channel (eV/Ch). This value can be adjusted based on the requirements. A larger value allows for a broader energy range to be collected, but the energy resolution will be lower. Conversely, a smaller value results in a narrower energy range but higher energy resolution, allowing for more detailed information to be obtained. The HAADF images of EELS mappings in Figure 3c and 3g were collected during EELS tests, and the contrast was kept high to get clear and distinct images, further evidencing that the samples could withstand the beam - under our experimental conditions- and that clear signals were obtained at the same time. Differently, once larger doses were applied as shown in Figure R2, MXenes were beam damaged, and acquiring HAADF images and EELS signals failed.

For all the EELS spectra, we only used the numerical filters to diminish the noise and didn't remove the background. As the thickness of our samples thinned by FIB is only tens of nanometers and the positions of the peaks are consistent with previous studies, Fourier-log deconvolution was not applied. More details were added in the Methods in the revised manuscript.

Fig. R2: (a) The HAADF image of lithiated HF-MXene after EELS tests. (b) The HAADF image of lithiated HF-MXene acquired during collecting EELS signals.

Changes to the manuscript:

1. Page 23, we have added the statement:

“The iDPC and HAADF images were conducted in spherical-aberration-corrected TEM instruments

(Spectra 300 X-CFEG (FEI) operated at 300 kV, and also ARM200F (JEOL) operated at 200 kV for comparison). The Beam currents were 0.05 nA and 0.01 nA. The convergence semi-angle was 25 mrad. The collection angle was 49 – 200 mrad and the dwell time was 2 μ s; The aberration coefficients were measured as: $A_1 = 2.54$ nm; $A_2 = 26$ nm; $B_2 = 19$ nm; $C_1 = 1.58$ nm; $C_3 = 469.8$ nm. EELS spectra were acquired on the Gatan 965 GIF QUANTUM ER with the GIF Continuum K3 camera at 200 kV. The energy dispersion of EELS-STEM is 0.45 eV Ch^{-1} . The samples for STEM imaging were prepared by an Thermo Scientific Scios 2 DualBeam Focused Ion Beam (FIB) system. 30 kV and 0.5 – 0.03 nA were applied to cut the samples into the slices. 5 kV and 48 pA were applied to thin the samples finely by FIB cutting. The samples were further cleaned by 2 kV and 43 pA. HRTEM and EDS characterizations of SEI layers were performed with an FEI Titan G 2 80-200 ChemiSTEM. EDS characterizations of Ga were performed with a Field Emission aberration-corrected Transmission Electron Microscope Hitachi HF5000 at 200 kV.”

Specific Comment 2: Sample Preparation

Sample preparation is critical for obtaining representative EM results, yet minimal information is provided. For example, if a FIB was used with Ga ions, these could interact with the MXenes. Was Ga detected in the EDX maps? Was low temperature used to minimize beam damage?

Response:

We thank the Referee for the important comments. We tried many sample preparation procedures and found that cutting the samples with low dose FIB to a thickness of less than 50 nm can provide the best sample quality and imaging resolution. To diminish the beam damage of the samples, 5 kV and 48 pA were applied to thin the samples finely by FIB cutting. The samples were further cleaned by 2 kV and 43 pA. Low temperature was not used. The EDS mapping and spectrum presented in Figure R3 below after FIB cutting - which has been added to the revised manuscript as Supplementary Figure 9 - shows that the Ga signal is quite low. Same sample preparation procedure was applied to prepare the MXene samples with SEI layer. The results clearly show the F signal, indicating that the SEI layer was well maintained. Thus, our experimental conditions are appropriate to maintain the structure of lithiated MXenes and SEI layer.

Fig. R3: The EDS mapping and spectrum of lithiated MS-MXene FIB milling sample.

Changes to the manuscript:

1. *Supplementary Figure 9 was added:*

Supplementary Fig. 9. The EDS mapping and spectrum of the region of lithiated MS-MXene FIB milling sample. The much weaker peaks of Ga compared to the characteristic peaks of MS-MXene (Ti, C, O and Cl) indicated the negligible effect of Ga ions on the samples.

2. *Page 12, we have added the statement:*

“The low magnification cross-section HAADF-STEM images of four lithiated MXene FIB samples are showed in Supplementary Fig. 8, exhibiting the layered structure. The EDS spectrum of lithiated MS-MXene FIB milling sample demonstrated the negligible effect of Gallium (Ga) ions on the samples during FIB cutting in Supplementary Fig. 9.”

Specific Comment 3: EELS Spectra

The EELS spectra in the supplementary information are insufficiently detailed:

i. The 50-70 eV range is not only the location of the Li K edge but also the Ti $M_{2,3}$ edge. Double plasmon effects might also be present. A broader energy range should be shown to assess the influence of the Ti edge or confirm if the bumps are just noise.

Response:

We thank the Referee for the insightful comments.

In the range from 10 to 70 eV, there are only three obvious peaks, as shown in Figure R4a. The two wider peaks at around 22 and 46 eV are the plasmon (Phys. Rev. B, 2003, 67, 165416), and the peak at around 63 eV represents Li (Matter, 2019, 1, 1232-1245). When compared with previous study (see Phys. Chem. Chem. Phys., 2018, 20, 25052-25061), as shown in Figure R4b, Ti $M_{2,3}$ edge peaks are located at around 40 and 47 eV, which don't overlap with the Li peak. We agree that there is a possibility Ti $M_{2,3}$ peaks in our results might be overlapped by the multiple plasmon excitation peaks. However, the scattering cross-section of multiple plasmonic effects decreases exponentially with increasing order ([Springer, 2011, 111-129], [Phys. Rev. B, 2003, 67, 165416]). Accordingly, the distinct peak at 63 eV is not considered to correspond to the multiple plasmon peak but more likely to the Li K-edge excitation.

Fig. R4: The Li EELS spectra of (a) our work and (b) published article (Phys. Chem. Chem. Phys., 2018, 20, 25052-25061).

ii. The authors do not clarify whether the spectra represent single pixels, pixel sums, or specific areas.

Response:

The EELS spectra represent single pixels in our work.

iii. How were the maps produced? How was the background in the Li-K region removed?

Response:

Regarding the mapping, we only performed numerical filter on the spectra with the same parameters and didn't remove the background.

iv. What does the spectrum at the $L_{2,3}$ edge of Ti in the gaps (Figure 3g) look like? The maps do not

convincingly show the absence of Ti, and the contrast scale could obscure its presence.

Response:

We provide Ti, C, Li and O spectra in different regions of lithiated MS- and HF-MXenes for comparison in Figure R5. Firstly, when comparing the global EELS spectra (orange region) of lithiated MS-MXenes and original MS-MXene, the Li peak is only visible for lithiated MS-MXenes. In addition, when we scan MXene Ti₃C₂ region only (blue frame in the STEM image), peaks characteristic of C and Ti are present but no signal of Li could be observed. When the interlayer region was scanned (green frame), Li signals could be observed and the Ti peak declined sharply in both MXenes as expected for clean interlayers. The Li signal was weak since the sample was quite thin and the amount of Li ions was not expected to be important (see electrochemical results as well). In addition, the C peak disappeared in MS-MXene interlayer, while C peak was present in HF-MXene interlayer, which supports the intercalation of carbonate solvents in HF-MXene together with Li ions. When comparing the C peak in the interlayers of HF-MXene and MS-MXene, the intensity of σ^* peak is much higher than π^* peak, and that also suggests the intercalation of carbonate solvents, in line with other studies ([Matter, 2021, 4, 1–11], [Matter, 2019, 1, 1232-1245], [Adv. Funct. Mater. 2022, 32, 2107190]). The Figure R5 was added as Supplementary Figure 13.

Fig. R5: EELS spectra of lithiated (a) MS-MXene and (b) HF-MXene. The spectra of the global region were marked as orange lines. The spectra of the interlayer regions were marked as green lines. The spectra of the MXene regions were marked as blue lines.

Changes to the manuscript:

1. *Supplementary Figure 13 were added:*

Supplementary Fig. 13. The EELS spectra of lithiated (a) MS-MXene and (b) HF-MXene. The spectra of the global regions were marked as orange lines. The spectra of the interlayer regions were marked as green lines. The spectra of the MXene regions were marked as blue lines.

Obvious peaks of Li illustrated the intercalation of Li ions [1-2]. When we scan MXene (blue frame) region only, characteristic peaks of Ti and C are obvious and similar to the pristine MXenes. Importantly, there is no signal corresponding to Li observed. When the scan region moved to the interlayer (green frame), Li signal could be observed and the Ti peak declined sharply in both MXenes as expected for clean interlayers. The Li signal was weak since the sample was quite thin and the amount of Li ions was not expected to be important (see electrochemical results as well). In addition, the C peak disappeared in MS-MXene while C peak maintained in HF-MXene, indicating the intercalation of carbonate solvents in HF-MXene. When compared the C peak in the interlayers of HF-MXene with C peak in HF-MXene, the intensity of σ^* peak are much higher (compared to the π^* peak), also verifying the intercalation of carbonate solvents, in line with other studies [3-5].

2. Page 15, we have added the statements:

“The signals corresponding to Li and O (and C) indicate Li ions and solvent molecules, respectively appear throughout the interlayers. The corresponding EELS spectra images also support the existence

of Li signals, and ion desolvation in MS-MXene and the ions-solvents co-intercalation in HF-MXene as shown in Supplementary Fig. 13.”

3. *References have been added in the Supplementary Information:*

[1] Zhang, W. *et al.* Kinetic pathways of ionic transport in fast-charging lithium titanate. *Science* **367**, 1030-1034 (2020).

[2] Saitoh, M. *et al.* Systematic analysis of electron energy-loss near-edge structures in Li-ion battery materials. *Phys. Chem. Chem. Phys.* **20**, 25052-25061 (2018).

[3] Zhang, Z. *et al.* Cathode-electrolyte interphase in lithium batteries revealed by cryogenic electron microscopy. *Matter* **4**, 302-312 (2021).

[4] Huang, W. *et al.* Dynamic structure and chemistry of the silicon solid-electrolyte interphase visualized by cryogenic electron microscopy. *Matter* **1**, 1232-1245 (2019).

[5] Yin, Z.-W. *et al.* Advanced electron energy loss spectroscopy for battery studies. *Adv. Funct. Mater.* **32**, 2107190 (2022).

Specific Comment 4: Resolution of EELS Spectra

i, The resolution (or zoom) of the EELS spectra is inadequate to confirm correspondence with typical MXene spectra (e.g., Figure 1g). The carbon K edge, in particular, does not resemble commonly reported results. Comparison with the literature is necessary. ii. To prove the presence of solvents in the interlayer gap, why did the authors not present the carbon K edge in this region? The fine structures of the carbon K edge differ significantly between carbonates (solvents) and MXenes.

Response:

We thank the Referee for the critical comments. The difference between our MXene EELS spectra and previously reported ones are attributed to variations in the sample thickness, which induces many changes including multiple scattering, energy losses and the ratio of surface/bulk contributions. To address this point, we adjusted the vertical axis to make the peaks more distinct in Figure R6. This Figure shows that the characteristic peaks of Ti (459 and 464 eV), C (286 and 290 eV) and O (532 eV) of our O- and Cl- terminated Ti₃C₂ are similar with the results of O-, F- and OH- terminated Ti₃C₂ previously reported in the literature (see Mater. Today Adv., 2021, 9, 100123).

According to reviewer's suggestion, we also added the carbon K-edge spectra to prove the

presence of solvents in the interlayer gap in Supplementary Figure 13.

Fig. R6: The EELS spectra of (a) pristine MS-MXene in our work and (b) pristine HF-MXene in published article (Mater. Today Adv., 2021, 9, 100123).

Changes to the manuscript:

1. *Figure 1* was revised:

Fig. 1 | The atomic structure of MS-Ti₃C₂T_x and HF-Ti₃C₂T_x. The HAADF-STEM images with low magnification of the FIB milling slices of (a) pristine MS-MXene and (b) pristine HF-MXene. The bright white ribbons marked by white arrows are the regions with the appropriate zone axis which could be imaged in iDPC-STEM mode. c The HAADF-STEM image of layer-structured MS-Ti₃C₂T_x with the *d*-spacing of 1.1 nm. Ti atoms are marked by blue dots. d The HAADF-STEM image of layer-structured HF-Ti₃C₂T_x with the *d*-spacing of 1.3 nm. Ti atoms are marked by blue dots. e The iDPC-STEM image of MS-Ti₃C₂T_x which is corresponding to (c). Ti atoms, C atoms and surface terminations are marked by blue, yellow and red dots, respectively. f The iDPC-STEM image of HF-Ti₃C₂T_x which is corresponding to (d). Ti atoms, C atoms and surface terminations are marked by blue, yellow and red dots, respectively. g The EELS spectra of MS-Ti₃C₂T_x indicating the existence of Ti, C, O and Cl signals. h HAADF-STEM image of the region terminated with -Cl in MS-Ti₃C₂T_x.

2. References have been added in the revised manuscript:

“Electron energy loss spectroscopy (EELS) analysis [36] was conducted on pristine MS-Ti₃C₂T_x, in which O signals (at around 530 eV) and weak Cl signals (at around 207 eV) were found in the spectra shown in Fig. 1g.”

[36] Alnoor, H. *et al.* Exploring MXenes and their MAX phase precursors by electron microscopy. *Mater. Today Adv.* **9**, 100123 (2021).

Specific Comment 5: iDPC Images

For Figure 3e: i. While iDPC interpretation is linked to atomic number (Z), it is highly sensitive to orientation and order. This sensitivity exceeds that of HAADF due to the composite processing of quadrant-detector signals. Thus, lower contrast in the interlayer gap does not necessarily indicate lighter elements. ii. Within the Ti₃C₂ slab, stripes and double spots in various areas suggest that the contrast is complex and prone to misinterpretation in disordered samples (e.g., Figure 1a). iii. The contrast scales for iDPC and HAADF (Figure 3e) should be consistent to determine if there is intensity in the gap in HAADF (it appears so). If true, this could indicate the presence of additional heavy atoms (in limited proportions) in the gap. iv. Were iDPC and HAADF images acquired simultaneously or sequentially? If sequentially, in what order?

Response:

We thank the Referee for the important comments.

Regarding comments *i* and *ii*: we fully agree that iDPC is highly sensitive to orientation and order of the samples. The samples being quite thin, if there is misorientation along the projection direction, we should get many additional spots within all of the Ti layers, C layers and functional group layers. We prepared more than ten micron-sized TEM samples and we checked almost all the areas in all these samples. In some cases, we obtained the image showing such misorientation, as shown in Figure R7 below. To ensure that our iDPC images were repeatable and reliable, we performed iDPC-STEM characterizations on MXenes with the appropriate zone axis. We compared the samples after and before lithiation, only the samples after lithiation show such weak contrast. Although it cannot be stated that the extra individual dots are all Li atoms (other atoms such as oxygen might be present), those results, combined with EELS mapping, ssNMR experiments, etc..., make it plausible to conclude that Li ions do insert in the interlayer spacing of MS-MXene without the solvents Li ions. We will make the huge database of image we collected accessible to the readers.

Regarding comments *iii* and *iv*: the iDPC and HAADF images were acquired simultaneously. We only performed HRTEM filter in DigitalMicrograph for the iDPC and HAADF images to make the images clearer and we didn't change the contrast or the brightness of the images.

Fig. R7: STEM images of MXene interlayers showing misorientation.

Specific Comment 6: Ordering of Solvent Molecules (Figure 3f and h)

The periodic contrast observed is unlikely to be due to Li ions (invisible under these conditions). It is more plausibly attributed to carbonate molecules. At room temperature, solvent molecules likely spin rapidly, making their ordering improbable. Under electron beam exposure, these molecules typically undergo radiolysis. Are the images repeatable? Do they depend on dose? The electron-conducting properties of MXenes might aid imaging of these sensitive molecules, but this needs to be verified and discussed.

Response:

We thank the Referee for insightful comments. We agree with the referee that periodic contrast observed in the interlayer of HF-MXene is unlikely to be due to Li ions, but more plausibly attributed to carbonate solvent molecules. The schematic showing the solvents co-intercalation in HF-Ti₃C₂T_x that we proposed in Figure 3h is based on the STEM images in Figure 3 and also on the XRD results from Figure 2. Then, the presence of Li in the interlayer spacing is supported by the NMR results (Figure 2e and f), and EELS mappings (Figure 3g). The fact that both Li ions and solvents can access the interlayers of HF-MXene and the solvents are located in the middle of the interlayers when the electrodes were polarized are also consistent with our MD simulations. We also agree with the comment on the random organization of the solvent molecules, but, as we only wanted to prove the solvents co-intercalation in HF-MXenes by using iDPC, EELS and other techniques, it was not necessary to clarify the orientation of solvent molecules in the interlayers.

As we previously mentioned in the answer to Comment 1, the quality of iDPC images being

dependent on the beam intensity, low electron doses were used to minimize the beam damage and we successfully conducted STEM characterizations of many samples using our determined experimental conditions detailed above, to ensure that our results were reproducible and reliable. To evidence that the electron beam intensity we used did not damage the samples, we actually compared samples analyzed with different doses, focusing on the thinner areas towards the end (the most sensitive zones to the electron beam) to ensure high-quality imaging under the right conditions, while the analysis of thicker areas was discussed before. As Figure R1 shows before – that we recall below – when a higher dose (0.1 nA) was applied, the MXene sample was damaged in few seconds while acquiring the image, resulting in blur and disordered contrast. In opposite, the contrast became weak and blur when a lower dose (5 pA) was applied. Notably, when 0.05 nA (corresponding to our experimental conditions) was applied, the atomic-scale structure and solvent layers were clearly imaged and the sample was not damaged. Thus, our determined experimental conditions are conducive to obtaining clear and repeatable iDPC images. Additionally, the conductivity (S/cm) of MXene ranges from thousands to tens of thousands ([Nature, 2014, 516, 78–81], [Nat. Rev. Chem., 2022, 6, 389–404]), which allows to alleviating the charge accumulation and acquiring the high quality iDPC images.

Fig. R1: The HAADF and iDPC images of lithiated HF-MXene samples with beam currents of (a) 0.1 nA, (b) 0.05 nA (our experimental condition) and (c) 5 pA.

Response to Referee Reports

Reviewer #4 (Remarks to the Author)

The authors have done a commendable job in complementing their manuscript, and I would like to thank them for their efforts. I still have a few comments (see below). However, in my opinion, they provide sufficiently strong evidence of their remarkable findings, along with enough details for others to reproduce them. Overall, their work meets the standards expected for publication in Nature Communications.

The authors' responses have largely addressed my concerns. Aside from minor revisions regarding the interpretation of π^ - σ^* features and the iDPC imaging discussion (where I trust the authors to make the necessary adjustments), I see no reason why this work should not be published in Nature Communications.*

General response: We would like to thank Referee#4 for their positive comments and recognition of our work, and we are glad to see that their concerns were addressed. In response to the new comments of the interpretation of π^* - σ^* features and the iDPC imaging discussion, we revised the EELS spectra and supplemented some statements. We provided point-by-point answers to each of the concerns raised. All the changes made to the revised manuscript have been highlighted in the main text.

Specific Comment 1: On beam damage and associated comments: The explanations are now much clearer and more convincing—thank you.

Response: We thank the Referee for their highly detailed, professional comments, which greatly contributed to improving our manuscript.

Specific Comment 2: On fine structures in EELS spectra: The dispersion of 0.45 eV/pixel accounts for the lack of clarity and the absence of subtle information in high-energy edges. The authors might consider using the DualEELS option in future studies (e.g., with 0.1 eV dispersion for two energy ranges) to gain deeper insights into chemical bonding, particularly in the interstitial gap. However, this goes beyond the scope of the present study.

Response: We thank the Referee for the insightful suggestions. We will definitively use the DualEELS option with the lower energy dispersion in future work, to explore the complex chemical bonding

nature at the atomic-scale in the future studies.

Specific Comment 3: *On the presence of π and σ peaks in MXene layers: I would reconsider the authors' addition suggesting the presence of π^* and σ^* peaks. Since carbon atoms are in an octahedral environment, π^* bands do not exist. The presence of two peaks and their evolution can certainly be discussed, but without assigning them a specific nature, as the interpretation is more complex than simply π^* (see studies on DFT band structures of $Ti_3C_2X_n$ compounds).*

Response: We thank the Referee for the insightful comments. We agree with the Referee that the first excitation peak in EELS spectra cannot be simply assigned to the π^* excitation. The excitation peak is certainly assigned to unoccupied states associated with the C p states, but these states are strongly hybridized with d states of Ti due to the local octahedral coordination field environment as mentioned by the Referee. Accordingly, in this revision, we focus more on the discussion of EELS spectra evolutions rather than explicit assignment of individual EELS excitation peaks. When comparing the C peaks in the interlayers of HF-MXene with C peaks in HF-MXene which are located at around 285 eV and 293 eV in Figure 1g and Supplementary Figure 13, despite the positions of the two carbon peaks remained basically unchanged, the peak at 293 eV in the interlayer was much broader with an extra shoulder in the range of 289 – 292 eV, as shown in Figure R1. It may be attributed to the carbonate peak integration (solvents) at around 290 eV according to previous work ([Matter, 2021, 4, 1-11], [Matter, 2019, 1, 1232-1245], [Adv. Funct. Mater. 2022, 32, 2107190]). We revised Supplementary Figure 13 and updated the corresponding statements in the revised Supplementary Information.

Fig. R1: The comparison of the carbon peaks of EELS spectra between HF-MXene and the HF-MXene interlayer. This Figure was added in Supplementary Figure 13.

Changes to the manuscript:

Supplementary Figure 13 was revised:

Supplementary Fig. 13. The EELS spectra of lithiated (a) MS-MXene and (b) HF-MXene. The spectra of the global regions were marked as orange lines. The spectra of the interlayer regions were marked as green lines. The spectra of the MXene regions were marked as blue lines.

Obvious peaks of Li illustrated the intercalation of Li ions. When we scan MXene (blue frame) region only, characteristic peaks of Ti and C are obvious and similar to the pristine MXenes. Importantly, there is no signal corresponding to Li observed. When the scan region moved to the interlayer (green frame), Li signal could be observed and the Ti peak declines sharply in both MXenes as expected for clean interlayers. The Li signal is weak since the sample is quite thin and the amount of Li ions is not expected to be important (see electrochemical results as well). In addition, the C peak

disappears in MS-MXene while C peak maintains in HF-MXene, indicating the intercalation of carbonate solvents in HF-MXene. When comparing the C peak in the interlayers of HF-MXene with C peak in HF-MXene, the peak at 293 eV in the interlayer is much wider with an extra shoulder in the range of 289 – 292 eV, as shown in the green region in the inset. It may be attributed to the carbonate peak integration at around 290 eV, also verifying the intercalation of carbonate solvents, in line with other studies.

Specific Comment 4: On EELS mapping (Comment 3iii): Typically, to produce EELS maps, the intensity in the relevant region (e.g., the Li K-edge) is extracted after background removal. I do not fully understand the response to my previous comment on this matter, but Figure R5 is quite informative. Since it has now been included as Supplementary Figure 13, this resolves my concern.

Response: We thank the Referee for the positive comments. We are happy to see that they appreciated the new data.

Specific Comment 5: On iDPC imaging: Thank you for the clarification. I particularly appreciate the response to Comment 5, where the authors adopt a more cautious approach and thoroughly discuss their conclusions based on these images. I agree that their overall experimental findings strongly support the presence of Li ions in the interlayer gap—this is particularly well supported by Supplementary Figure 13. Regarding the iDPC images, I believe it would be appropriate to use the term “plausible” in the manuscript. Since a titanium signal is detected in the interlayer region (Supplementary Figure 13), these atoms may also appear in the iDPC images of this area. I also appreciate the authors making their image database available to readers.

Response: Thank you for your constructive comments on the iDPC characterization, which really improved the reliability of our results. We agree that a more cautious approach should be adopted since not all the individual dots are Li ions. As suggested, we added the term “plausible” and updated the statements in the revised manuscript for the rigorous expression.

Changes to the manuscript:

Page 11, we updated the statement in the revised manuscript:

“Moreover, intercalated Li ions are clearly shown in the iDPC-STEM images while they are absent in the pristine MS-Ti₃C₂T_x (Fig. 1e). It is worth noting that although the extra dots in the interlayer

spacing in Figs. 3a and b may not all correspond to lithium atoms individually, **the majority of the contrast likely originating from Li ions is plausible** since it is quite weak and only appear in iDPC-STEM mode. In Fig. 3c and Supplementary Fig. 11a...”

Specific Comment 6: For the ordering of molecules, while this is not the main result and is therefore not critical for acceptance, a non-specialist in STEM imaging techniques might misinterpret the findings as proof of molecular ordering, which has not been demonstrated (as acknowledged by the authors). It would be helpful to clarify this point further.

Response: We thank the Referee for the important comments. The point has been clarified as suggested in the revised manuscript, as shown below.

Changes to the manuscript:

Page 13, we supplemented the statement in the revised manuscript:

“This process is then different from that of MS-Ti₃C₂T_x MXene (T_x = –O and –Cl) where fully desolvated Li ion intercalation is observed (Fig. 3d). **It should be noted that the schematic does not indicate the real orientation and amount of the solvent molecules here.** Since the main difference between the two MXene samples is the surface terminations, this suggests that the surface chemistry drives the Li ion desolvation and the charge storage mechanism.”